# Mechanism for microbial population collapse in a fluctuating resource environment

Serdar Turkarslan[1] iD, Arjun V Raman[1], Anne W Thompson[1], Christina E Arens[1], Mark A Gillespie[1], Frederick von Netzer[2], Kristina L Hillesland[3], Sergey Stolyar[1], Adrian López García de Lomana[1], David J Reiss[1], Drew Gorman-Lewis[4], Grant M Zane[5], Jeffrey A Ranish[1], Judy D Wall[5], David A Stahl[2] & Nitin S Baliga[1,*] iD

## Abstract

Managing trade-offs through gene regulation is believed to confer resilience to a microbial community in a fluctuating resource environment. To investigate this hypothesis, we imposed a fluctuating environment that required the sulfate-reducer *Desulfovibrio vulgaris* to undergo repeated ecologically relevant shifts between retaining metabolic independence (active capacity for sulfate respiration) and becoming metabolically specialized to a mutualistic association with the hydrogen-consuming *Methanococcus maripaludis*. Strikingly, the microbial community became progressively less proficient at restoring the environmentally relevant physiological state after each perturbation and most cultures collapsed within 3–7 shifts. Counterintuitively, the collapse phenomenon was prevented by a single regulatory mutation. We have characterized the mechanism for collapse by conducting RNA-seq analysis, proteomics, microcalorimetry, and single-cell transcriptome analysis. We demonstrate that the collapse was caused by conditional gene regulation, which drove precipitous decline in intracellular abundance of essential transcripts and proteins, imposing greater energetic burden of regulation to restore function in a fluctuating environment.

**Keywords** fluctuating resource environment; microbial population collapse; regulation; resilience; syntrophy
**Subject Categories** Microbiology, Virology & Host Pathogen Interaction; Quantitative Biology & Dynamical Systems
**Mol Syst Biol.** (2017) 13: 919

## Introduction

Generalist organisms can adapt to many environments by up- and downregulating genes to tailor their physiology to each environment (Futuyma & Moreno, 1988). Not surprisingly, experimental evolution of a generalist with a single resource improves growth characteristics in that environment but at the cost of diminished capability to adapt to other resources; that is, they become more specialized (Cooper & Lenski, 2000; Blount *et al*, 2008; Lee *et al*, 2009; Hillesland *et al*, 2014). Many of these studies have also made the observation that improved growth characteristics correlate with appearance of mutations in regulatory elements (Kurlandzka *et al*, 1991; Cooper *et al*, 2003; Barrick *et al*, 2009; Yang *et al*, 2011; Hindré *et al*, 2012; Hottes *et al*, 2013). The mechanism by which mutations in a gene regulatory network (GRN) might positively impact the fitness of a species in fluctuating environments is, however, unclear. Modeling and analysis of microbial genomes suggests fluctuating environments select for greater independence of regulatory networks for individual functions (Kashtan & Alon, 2005; Parter *et al*, 2007). Mutations that disrupt precise regulation of these modular GRNs could impede efficient acclimation with each environmental fluctuation. Thus, generalism may evolve through the fine-tuning of regulation of modules. On the other hand, specialization seems to evolve through loss-of-function mutations in the GRN, which may remove unnecessary functions and optimize others (Hottes *et al*, 2013). In this context, resilience can be defined as the property of a generalist to dynamically adapt to changing environmental conditions by appropriately fine-tuning relevant physiological capabilities to optimally use resources in its environment.

In this study, we have applied a systems approach to characterize the resilience of a generalist in a fluctuating resource environment in which it is required to dynamically switch between living independently or in a mutualistic interaction with a second organism. This two-organism model community was previously established under sulfate-deplete conditions that prevented independent growth of *Desulfovibrio vulgaris* (Dv) by sulfate respiration (SR), and required its syntrophic interaction with *Methanococcus maripaludis* (Mm) to support growth of both organisms (Hillesland & Stahl, 2010). "Syntrophy" (ST) is an obligate mutualism in which

1 Institute for Systems Biology, Seattle, WA, USA
2 Civil and Environmental Engineering, University of Washington, Seattle, WA, USA
3 Biological Sciences, University of Washington Bothell, Bothell, WA, USA
4 Earth and Space Sciences, University of Washington, Seattle, WA, USA
5 Department of Biochemistry, University of Missouri, Columbia, MO, USA
*Corresponding author. Tel: +1 206 732 1266; E-mail: nbaliga@systemsbiology.org

the oxidation of the organic substrate (lactate) by Dv is energetically feasible only if its fermentation products (primarily hydrogen ($H_2$) and formate) are consumed by Mm. Hillesland and Stahl (2010) further subjected twelve replicate cultures of this model community to 1,000 generations of experimental evolution and observed significantly improved growth characteristics across all 12 lines around 300 generations (Hillesland & Stahl, 2010). Whole-genome sequencing revealed that many mutations had accumulated in the genomes of both organisms across all 12 replicate cultures (Table EV1). Analysis of mutations showed that specialization for ST across nearly all lines came at the expense (tradeoff) of erosion of SR (Hillesland *et al*, 2014). Thus, the work of (Hillesland & Stahl, 2010; Hillesland *et al*, 2014) investigated physiological and genomic changes across Dv and Mm when the two organisms were required to co-evolve in an obligately interdependent syntrophic association. The major finding from this prior work was that the obligate association with Mm resulted in erosion of sulfate respiration functions, thereby compromising the capability of Dv to live independently if excess sulfate were to become available in the future.

Here, we have conducted further analysis of the mutational data from the prior study to discover that in addition to erosion of SR, mutations had also accumulated at high frequency in components of the regulatory network—a finding that raised interesting questions regarding the role of regulation in evolution. We have investigated this question in great depth through inference of the gene regulatory network to identify regulators of SR, construction of regulatory mutants; laboratory evolution of wild-type and regulatory mutant co-cultures in *fluctuating* environmental conditions, and extensive molecular, transcriptomic, proteomic, and single-cell characterization. Through this work, we have uncovered the importance of regulation and regulatory mutations on resilience of a microbial community in a fluctuating resource environment.

# Results and Discussion

## Regulatory functions accumulate mutations during laboratory evolution of obligate ST

We expanded analysis of sequencing data from the Hillesland *et al* (2014) study to examine frequencies, genomic locations, and functional categories of mutations that had accumulated during experimental evolution of obligate ST between a Dv and Mm two-member microbial community. We observed a significant enrichment of mutations in coding regions of regulatory genes and also within intergenic regions in Dv (Fig 1A and B), potentially affecting both trans- and cis-regulation of downstream genes. The functional categories for all mutated genes were enriched for regulation of transcription (GO:0045449, *P*-value: 0.00632), transcription regulator activity (GO:0030528, *P*-value: 0.05570), and two-component sensor activity (GO:0000155, *P*-value: 0.00608; Table EV2). Specifically, across the 12 lines, we found 112 mutations within coding regions of 33 regulatory genes (Biological regulation, GO:0065007, *P*-value: 0.01018) and 365 mutations mapping to 108 unique intergenic loci ($P$-value $< 10^{-4}$). It is tempting to speculate from these results that some aspects of gene regulation in Dv must have been dispensable because of the limited physiological

adjustments required for sustaining obligate ST. For instance, some of the intergenic mutations were proximal to genes such as two lactate permeases, and Ech hydrogenase—functions associated with SR, which was ultimately lost across most lines. Based on this result, we speculated that disruptive mutations in the GRN for SR would be detrimental in a variable environment with fluctuating availability of sulfate, which should favor an intact GRN that can optimize sulfate utilization by appropriately up- or downregulating SR.

## GRN model identifies regulators that enable adaptation to an upshift in sulfate availability

Since regulators of SR were unknown, we used a systems biology approach to discover them by deciphering the GRN of Dv. Using a large compendium of Dv transcriptional response to diverse environmental factors (684 microarrays interrogating transcriptional responses to 25 unique perturbations) along with genomic sequence data (Dehal *et al*, 2009), we reconstructed the first global and predictive model of the GRN governing environmental responses of Dv (Table 1, Materials and Methods, Table EV3 and Table EV11; Reiss *et al*, 2006; Bonneau *et al*, 2007). The Environment and Gene Regulatory Influence Network (EGRIN) model organized 2,984 of 3491 (~85%) genes in the Dv genome into 170 regulatory modules and modeled their regulation by 122 transcription factors and 12 environmental factors. The EGRIN model made accurate predictions of transcriptional responses to new environments. The root-mean-square deviation (RMSD) of predicted transcriptional changes for all modules was similar over the training data set (RMSD = 0.42) and a new dataset that was not used in model construction (RMSD = 0.41; Bonneau *et al*, 2006). We further verified the EGRIN model by demonstrating that 38 gene modules within EGRIN had accurately recapitulated 31 out of 77 evolutionarily conserved Dv regulons curated in RegPrecise v4 (Novichkov *et al*, 2013) covering 17 transcription factor families and 1 RNA regulator (BH-corrected *P*-value < 0.005, Table EV4). In addition to reconstructing regulon memberships, EGRIN predicted the cognate transcription factor for each regulon and its binding location within individual gene promoters (Table EV3, Model EV1, the EGRIN model for Dv can be explored at http://networks.systemsbiology.net/syntrophy).

Transcription factors in the EGRIN model were rank ordered by the predicted weight of their influence on regulatory modules that were enriched for SR genes. Altogether three transcription factors were predicted to regulate key SR genes across four modules, including two sigma-54 family transcription factors: DVU0744 (a repressor of 128 target genes across nine modules, of which two contained SR genes) and DVU2275 (an activator of 119 target genes across eight modules, of which three contained SR genes), and a GntR family transcription factor: DVU2802 (a repressor of 240 target genes across 16 modules, of which three contained SR genes; Fig 2A and Table EV3). We characterized the predicted roles of these three transcription factors by assaying SR-relevant fitness effects of transposon-insertions in each of the three genes. Surprisingly, none of the three transcription factor mutants had a phenotype (measured as growth rate) under SR conditions with excess sulfate in the growth medium. By contrast, fitness of all three Dv mutants was significantly reduced when they were transitioned from a sulfate-depleted condition to a growth medium with excess sulfate,

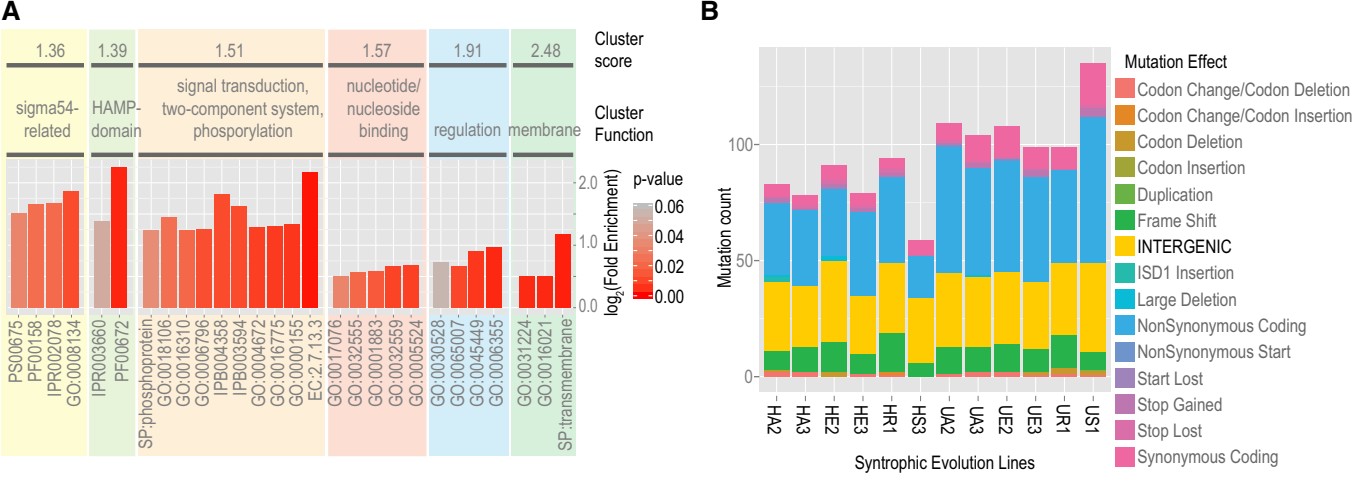

**Figure 1.  Mutations accumulate in the gene regulatory network during laboratory evolution of syntrophy.**

A   Functional enrichment of mutations accumulated during experimental evolution of Dv-Mm syntrophic community. Fold-enrichment for enriched ontology terms (*P*-values < 0.05) is shown in functionally related clusters.

B   Number, type, and SnpEff-predicted effects of mutations (Cingolani *et al*, 2012) accumulated over 1,000 generations during evolution of obligate ST between Dv and Mm. Enrichment of intergenic mutations was determined by a permutation test.

supporting the hypothesis that regulation of SR is important only in response to a change in sulfate availability (Fig 2B and C; Alon, 2007).

### Dv regulatory mutants establish stable co-cultures with Mm during repetitive shifts between SR and ST, by contrast wild-type lines reproducibly collapse

We examined growth characteristics of the wild-type strain and the three mutants when growing with Mm in dynamically changing resource environments that support SR or ST at different times. Six replicate co-cultures with Mm were established for each of the four

**Table 1.  Summary of EGRIN model.**

| cMonkey model | | Inferelator model | |
|---|---|---|---|
| Genes | 2,984 of 3,491 total genes in the genome | Transcription factors | 128 |
| Conditions | 684 | Environmental factors | 16 |
| Regulatory modules | 170[a]/349 | | |
| Motifs | 148[b]/662 | | |
| Network[c] | | | |
| Nodes | 306[d] | Upregulation | 573 |
| Edges | 919 | Downregulation | 346 |

Environment and Gene Regulatory Influence Network (EGRIN) model was constructed as described in Materials and Methods. This model includes both cMonkey and Inferelator algorithm input data and results together with different filtering parameters as defined below.
[a] Number of modules filtered for residual ≤ 0.5.
[b]Number of motifs filtered for *e*-value < 10.
[c]Network is filtered to only include influences with influence weight < −0.1 or > +0.1 and module residual < 0.5.
[d]Nodes include regulators and their targets regulatory modules.

strains in a sulfate-depleted growth medium. The co-cultures were then subjected to transitions between a medium containing lactate and sulfate or just lactate, requiring Dv to repeatedly switch between SR and Mm-dependent ST, respectively—physiological adjustments that are ecologically relevant. Based on results in Fig 2B and C, we expected that the regulatory mutants would be ineffective at driving environment-responsive transitions between ST and SR physiological states. Contrary to our expectation, the wild-type co-cultures rapidly and reproducibly collapsed within seven transfers, across all replicate lines. Strikingly, two of the three mutants sustained facultative ST over a longer time frame, with DVU0744::Tn5 never collapsing over 24 transitions (Fig 3). In contrast, only two of 24 replicate co-cultures collapsed over a similar time scale in the obligate ST evolution experiment (Hillesland & Stahl, 2010).

### Collapse of wild-type co-culture lines results from unsustainable conditional regulation in a fluctuating environment

We explored two potential mechanisms that might have led to population collapse of the wild-type co-cultures, but sustained the DVU0744::Tn5 synthetic community.

*Fast-growing wild-type Dv outgrew Mm*
The rapid adaptation of the wild-type strain to each shift (Fig 3) might have progressively resulted in serial dilution of Mm cells to a point that ST was unsustainable. The slower growth of DVU0744::Tn5 post-transition to SR conditions might have maintained higher numbers of Mm cells and, therefore, circumvented collapse. However, collapse across most lines occurred under SR conditions that should have favored the wild type. Notwithstanding this observation, we counted the number of Dv and Mm cells through the course of the experiment using two independent techniques: microscopy and flow cytometry. There was no significant difference in

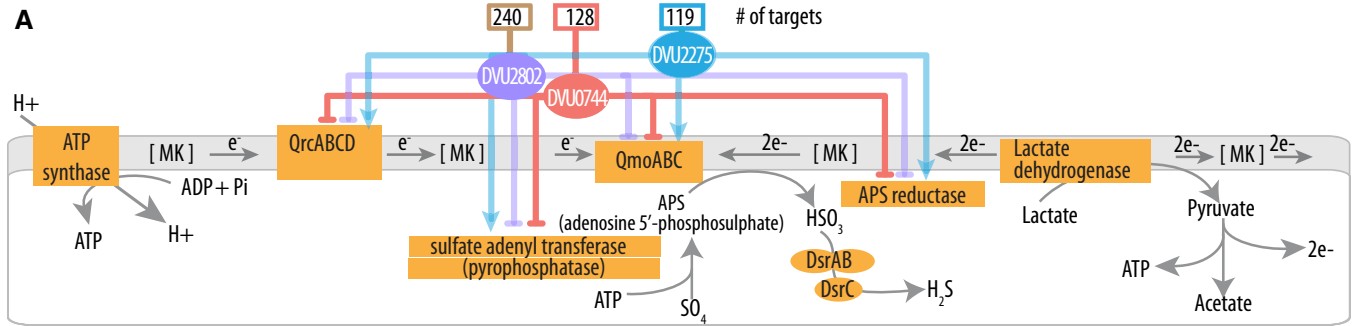

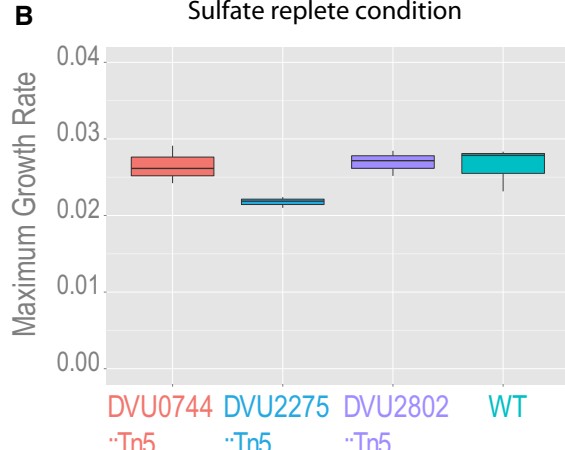
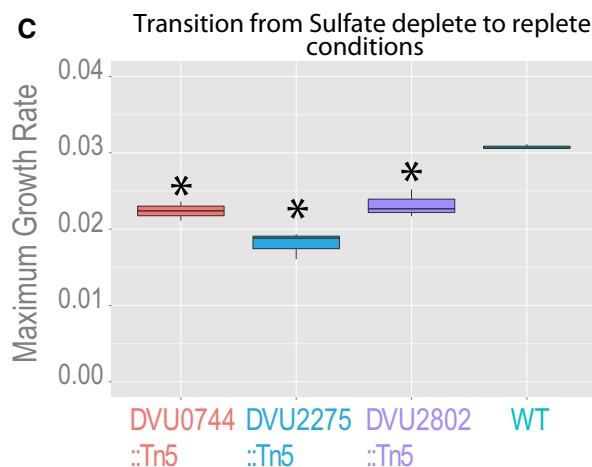

**Figure 2. Sulfate respiration regulators identified using systems approach are necessary for transitioning to conditions with increased sulfate availability.**

A   Three transcriptional regulators were predicted by the EGRIN model to influence the expression of SR genes. Influence of each regulator on its predicted targets is shown with color-coded lines, with total number of gene targets above each regulator.

B   Maximum growth rates for all Dv strains in excess sulfate condition.

C   Maximum growth rates for wild type and the three regulatory mutants of Dv subsequent to transfer from a sulfate-depleted condition to a growth medium with excess sulfate. Significant growth rate differences relative to wild type are indicated with an asterisk.

Data information: (B and C) The lower and upper ends of the boxes ("hinges") correspond to the first and third quartiles (the 25th and 75th percentiles). Horizontal lines correspond to median values. Error bars extend from the upper or lower hinges to the highest or lowest values that are within 1.5× IQR (interquartile range) of the hinge, respectively. Measurements are from three replicates.

Source data are available online for this figure.

relative ratios of Dv:Mm cells throughout the experiment, across both the wild-type and DVU0744::Tn5 lines. Thus, we could rule out that population collapse in the co-culture with wild-type Dv might have resulted from dilution of Mm cells (Figs 4 and EV1).

*Conditional regulation in a continually fluctuating environment is unsustainable*

Gene regulation minimizes the cost of cellular operations by turning up or down physiological processes in an as-needed manner depending on resource availability (Nagarajan *et al*, 2013). While this justifies adaptive value of gene regulation in a variable resource environment, regulatory systems can become burdensome and even a bane in an environment that seldom changes or one that changes too frequently (Alon, 2007). We analyzed global transcriptome changes in wild-type Dv and DVU0744::Tn5 during ST and SR growth modes, and following transitions between the two (Fig EV2). Over 1,300 genes were differentially regulated in the wild type demonstrating extensive gene regulation during various phases

of the experiment, especially upon transitioning between SR and ST conditions (Fig 5A, upper panel). This result is comparable to the reported differential expression of 1,202 genes upon switching from ST to sulfidogenic lifestyles of Dv co-cultured with *Methanosarcina barkeri* (Plugge *et al*, 2010). A significant fraction of the differentially regulated genes were determined to be essential (essentiality is determined based on Rapid Transposon Liquid Enrichment Sequencing and an associated model for essentiality (Fels *et al*, 2013) for growth by SR (Fig 5A, lower panel; 264/1,300; *P*-value: $8.1 \times 10^{-8}$, Table EV5; Fels *et al*, 2013).

Regulation of more than 80% of the genes (1,148 genes) that changed in the wild type was disrupted or altered in DVU0744::Tn5, demonstrating an important role for DVU0744 in globally coordinating transcription during transitions between SR and ST (Table EV5). The system-wide consequences of knocking out DVU0744 could be attributed to its predicted regulation of five signal transduction genes and seven transcriptional regulators; expression of four of these regulatory genes was perturbed in DVU0744::Tn5. Affected

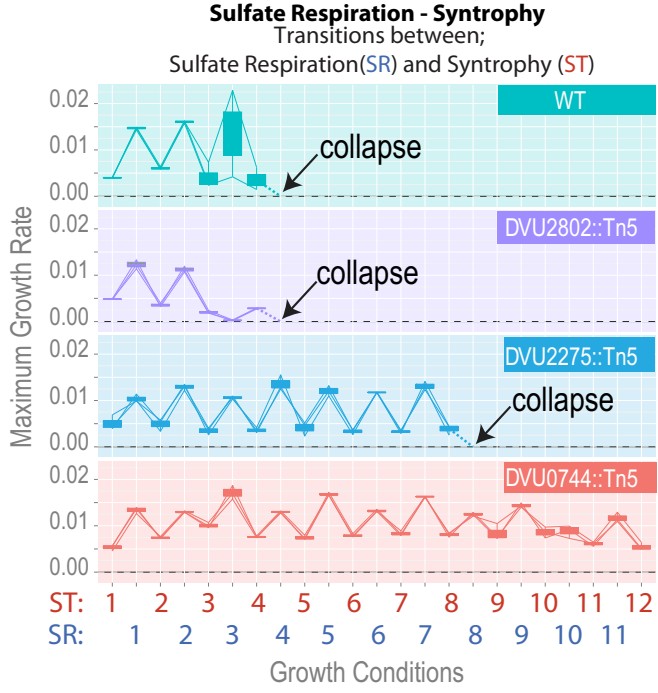

**Figure 3.  Regulatory mutations delay or prevent population collapse.**
Co-cultures of wild-type Mm were established with each of the three regulatory mutants or wild-type Dv under ST conditions. Each of the four co-cultures was alternated between growth conditions that favor SR or ST by transferring an aliquot from an early log phase co-culture into the next growth medium. Growth rates for co-cultures across conditions supporting SR or ST are shown. Numbers indicate *n*-th transition for either SR or ST conditions.

Source data are available online for this figure.

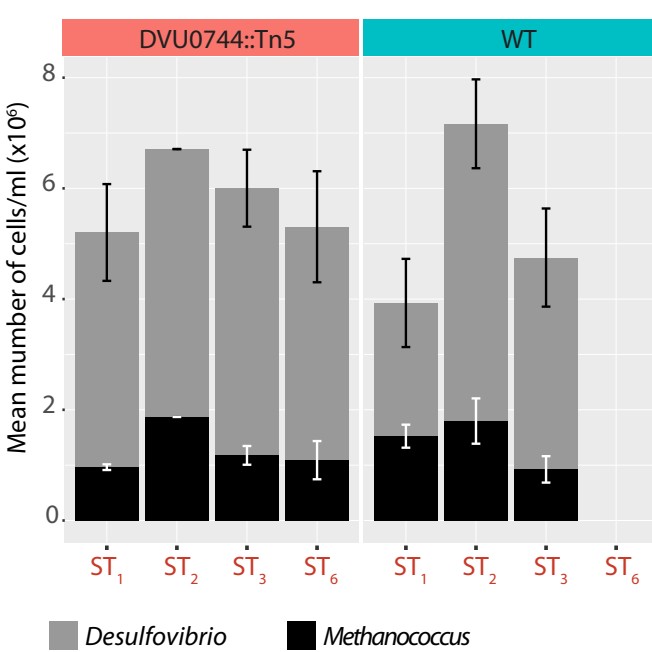

**Figure 4.  Relative ratios of Dv:Mm cells do not change in wild type and DVU0744::Tn5 through the laboratory evolution experiment.**
Hemocytometer counts of Dv and Mm cells in wild-type and DVU0744::Tn5 co-cultures across ST conditions. We determined cell numbers by using both microscopy and flow cytometry in order to rule out that population collapse in the co-culture with wild-type Dv might have resulted from washing out of Mm cells. Similar results were observed with flow cytometer counts (Fig EV1). Error bars correspond to standard deviation across 3 replicate measurements.

Source data are available online for this figure.

genes in DVU0744::Tn5 were enriched for functions in two-component signal transduction (52; *P*-value:$1.1 \times 10^{-6}$), regulation of transcription (64; *P*-value: $1.8 \times 10^{-5}$), and bacterial chemotaxis (11; *P*-value: $1.9 \times 10^{-5}$; Table EV6) –functions that have been demonstrated to be important for ST in other species (Kosaka *et al*, 2008; Kato *et al*, 2009; Shimoyama *et al*, 2009). Twenty-seven affected genes were predicted targets of DVU0744 in the EGRIN model, 141 were deemed essential for SR (Fels *et al*, 2013), 70 accumulated mutations (including missense, nonsense, and frameshift mutations) during experimental evolution of obligate ST (Hillesland *et al*, 2014), and 93 have been previously shown to be important for ST (Walker *et al*, 2009).

**Model predicts that conditional repression drives dilution of essential proteins within individual cells**

To understand the consequence(s) of disrupted regulation of SR and elucidate the mechanism for population collapse, we adapted the model for the lac system developed by Cai *et al* (2006). We selected this model because like the lactose-dependent repressor LacI, DVU0744 is a transcriptional repressor that is sensitive to sulfate concentration, which is evidenced by the growth impairment during transitions from sulfate-deplete to sulfate-replete conditions (Fig 2C). We used the model to investigate how disrupted regulation would affect the cellular protein copy number

distribution in a rapidly fluctuating environment. More specifically, we were interested in understanding how loss of condition-specific repression of essential genes during ST growth affected their protein copy number.

Analogous to the lac system, the model is developed on the assumption that presence or absence of sulfate can lead to an all-or-none type induction of SR-essential genes. Applying this model to the experimental evolution regime predicted that conditionally repressed genes are dramatically diluted due to iterative transitioning between SR and ST conditions. Specifically, with DVU0744 present in the wild-type strain, all-or-none type induction is probable for many essential genes and thus most cells will transition to a state that is appropriate for the environmental condition—a property termed "*relational resilience*" (Song *et al*, 2015). Conversely, with repetitive shifting of the environmental condition, most wild-type cells lose conditionally essential proteins due to complete repression of genes relevant for the opposite growth condition. With the absence of DVU0744 in the mutant strain, complete repression is not achieved, and leaky expression of normally repressed genes confers a hybrid state that is tolerant of repetitive transition between SR and ST, albeit with a fitness cost associated with a single transition (Fig 2C). We found with this model that the population of wild-type cells rapidly loses essential proteins that are repressed in ST, but not fully restored during SR (Fig 5B). The mutant strain also experiences a decline in the fraction of cells with nonzero copies of the protein, but it is not as dramatic. Thus, loss of repression in the

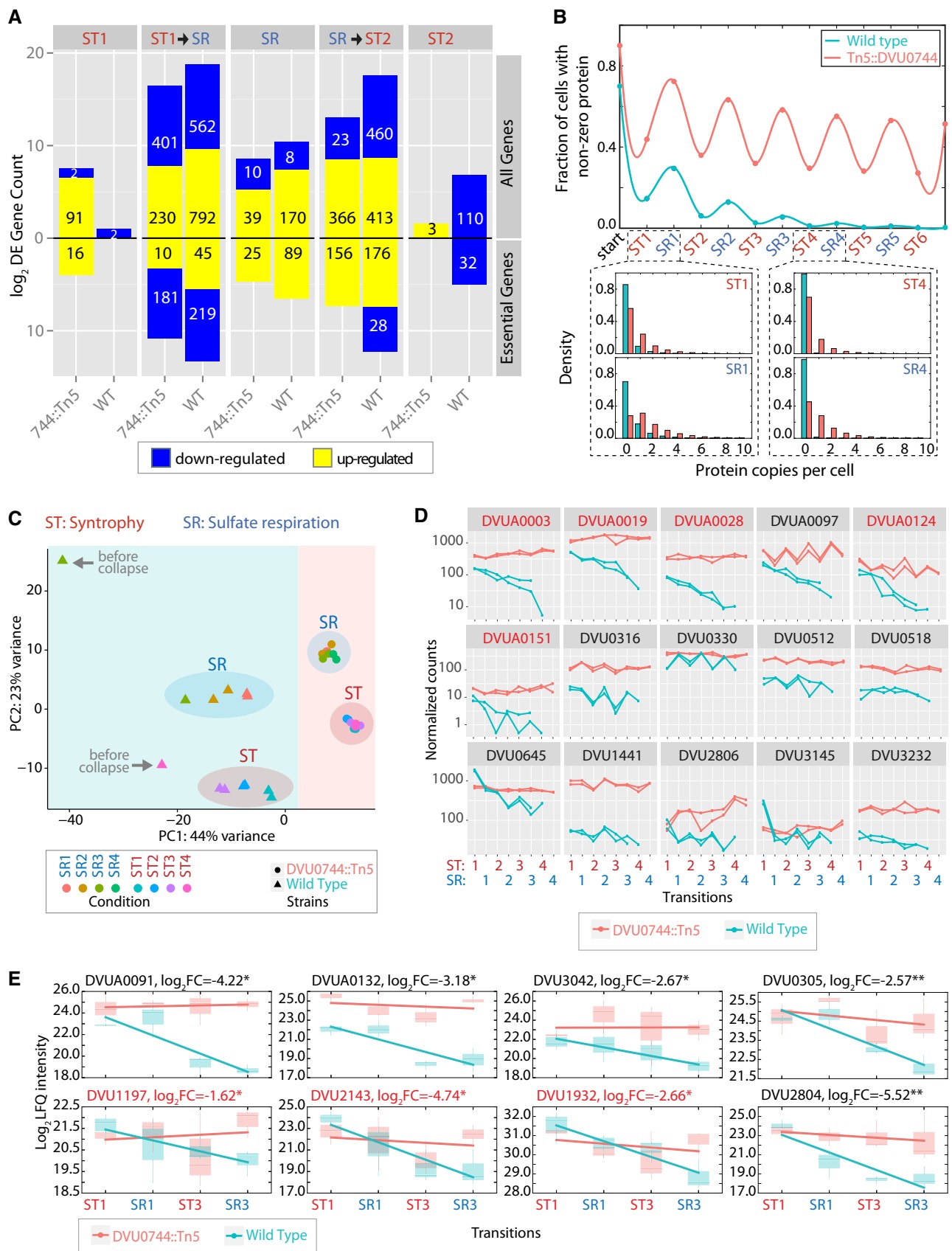

**Figure 5.**

◀

**Figure 5.  Disruption of conditional regulation in mutant prevents dilution of transcripts and proteins in fluctuating environments.**

A   Global transcriptomes were profiled and compared between early and mid-log phase within each condition, denoted ST1 (first iteration of ST), SR, and ST2 (second iteration of ST); and after transition between ST and SR. Upward barplot shows all differentially expressed genes, and the downward barplot shows only essential genes.

B   We simulated the protein copy number distribution in each phase of growth (ST and SR), assuming that a hypothetical gene was repressed in ST, but essential for SR. The fraction of cells with nonzero copies of an essential protein for sulfate respiration is plotted over several transfers (upper panel). Lower panel shows histogram of protein copy numbers per cell for selected early (ST1 and SR1) and later (ST4 and SR4) transitions.

C   PCA plot created by DESeq2 R package for visualizing clustering of experimental covariates based on the normalized read counts.

D   Change in abundance of selected Dv transcripts during experimental evolution of wild-type and mutant co-cultures. Normalized RNA-seq read counts for each gene across two replicates for wild type (green) and mutant (red) are plotted on a $\log_{10}$ scale. Essential gene names are indicated in red. See Appendix Fig S1 for the complete list of transcripts.

E   $\log_2$-fold change of protein abundance in ST1, SR1, ST3, and SR3 conditions, in the wild-type and mutant co-cultures (* indicates *P*-value < 0.05, or ** indicates *P*-value < 0.01). Essential gene names are indicated in red. See Table EV10 for the complete list of proteins. The lower and upper ends of the boxes ("hinges") correspond to the first and third quartiles (the 25th and 75th percentiles). Horizontal lines correspond to median values. Error bars extend from the upper or lower hinges to the highest or lowest values that are within 1.5× IQR (interquartile range) of the hinge, respectively.

Source data are available online for this figure.

mutant strain maintains a higher proportion of cells (relative to the wild type) with at least one copy of the regulated protein. This supports the notion that conditionally repressed, essential proteins are progressively lost causing population collapse of the wild-type co-culture. We can now posit that this phenomenon is generalizable to other systems including the lac system and adaptive evolution of *E. coli* to fluctuations in lactose concentration, a set up on which the Cai *et al* model was based.

**Predicted dilution due to conditional regulation is confirmed at the level of transcripts, proteins, and metabolic activity**

We looked for evidence of the model-predicted dilution effect at the level of transcripts, proteins, and metabolites.

*Evidence for dilution at the transcript level*
We used RNA-sequencing to globally track changes in transcript abundances in wild type and DVU0744::Tn5 during repeated transitions of their respective co-cultures between ST and SR conditions, up until (wild type) and beyond (DVU0744::Tn5) the point of collapse. First, we performed principal component analysis (PCA) of the top 500 most variant genes across the entire experiment, in order to assess overall similarity and differences between samples. The PCA showed that transcriptomes of the wild type and mutant could be distinctly attributed to the genotype (PC1) and ST/SR physiological states (PC2; Fig 5C). Interestingly, the regulatory mutant reproducibly restored its original ST or SR state subsequent to each transition, whereas the wild type was unable to do so, with each transition pushing it farther from the original ST and SR states. The transcriptome of the wild type appeared dramatically different than the appropriate state for the growth condition just prior to the population collapse, suggesting that inability to adopt the appropriate (SR or ST) attractor state (i.e. loss of *relational resilience* (Song *et al*, 2015) is an early warning sign for population collapse. Furthermore, consistent with the model prediction, we observed progressive dilution of transcripts in wild type relative to the mutant. Approximately, 146 transcripts, including eight essential genes, suffered steady decline in abundance over progressive transitions in wild type but not in the regulatory mutant (Fig 5D and Appendix Fig S1). A significant number (118 genes) of affected genes were plasmid-encoded—while this observation warrants further investigation that is outside the scope

of this study, it is noteworthy that many of these plasmid-encoded genes are essential. Taken together all affected transcripts were enriched for bacterial type III secretion and flagella-related pathways and also included many energy production pathway-related genes, including hydrogenase.

*Evidence for dilution at the protein level*
We performed quantitative, shotgun proteomics to track abundance changes in 728 proteins during transitions of Mm co-cultures with wild type and DVU0744::Tn5 between SR and ST (Table EV10, Materials and Methods). Using linear regression, we discovered that at least 52 proteins were progressively diluted in the wild-type background, but not in the regulatory mutant (Fig 5E and Appendix Fig S2). The likelihood of this trend was determined to be statistically very significant (*P*-value: 0.0025). Functional roles for these proteins included signaling, transport, and amino acid metabolism, and eight had been previously determined to be essential for survival in SR conditions. Notably, there were only three proteins with a reverse trend; that is, they were diluted in the mutant but not in the wild type—demonstrating unequivocally that regulation by DVU0744 in response to frequent transitions between SR and ST leads to progressive dilution of both transcripts and proteins.

*Evidence for dilution at the metabolite level*
Finally, we monitored metabolic activity of Dv across transfers between ST and SR in a new set of laboratory evolution experiments. Through successive transfers, rapid decline in lactate consumption and $H_2$ evolution (proxies for metabolic activity of Dv) supported the model-predicted dilution of proteins performing essential biochemical functions. Consistent with the model prediction, DVU0744::Tn5 achieved steady-state levels of $H_2$ evolution (Fig 6A) and consumption of lactate (Fig EV3).

**Disrupted regulation lowers energetic burden for adaptation to a fluctuating environment**

The drift in the transcriptional state of the wild type from the original state for SR and ST suggested that it would have to expend greater energy than the regulatory mutant to restore its physiological state for the new growth condition. The energetic cost of regulation in the wild-type and regulatory mutant co-cultures was determined using microcalorimetry to measure heat production

following transitions into SR in the beginning and at a later stage of the laboratory evolution experiment (von Stockar & Liu, 1999; Chardin *et al*, 2003; von Stockar *et al*, 2006). We used total amount of heat released as a proxy for total energy expenditure of cells in

each culture, and attributed the difference in heat produced by wild type versus mutant to lack of regulation in the latter. Our rationale was that both strains would have comparable energy terms for growth, maintenance, etc., and that the only difference would be in

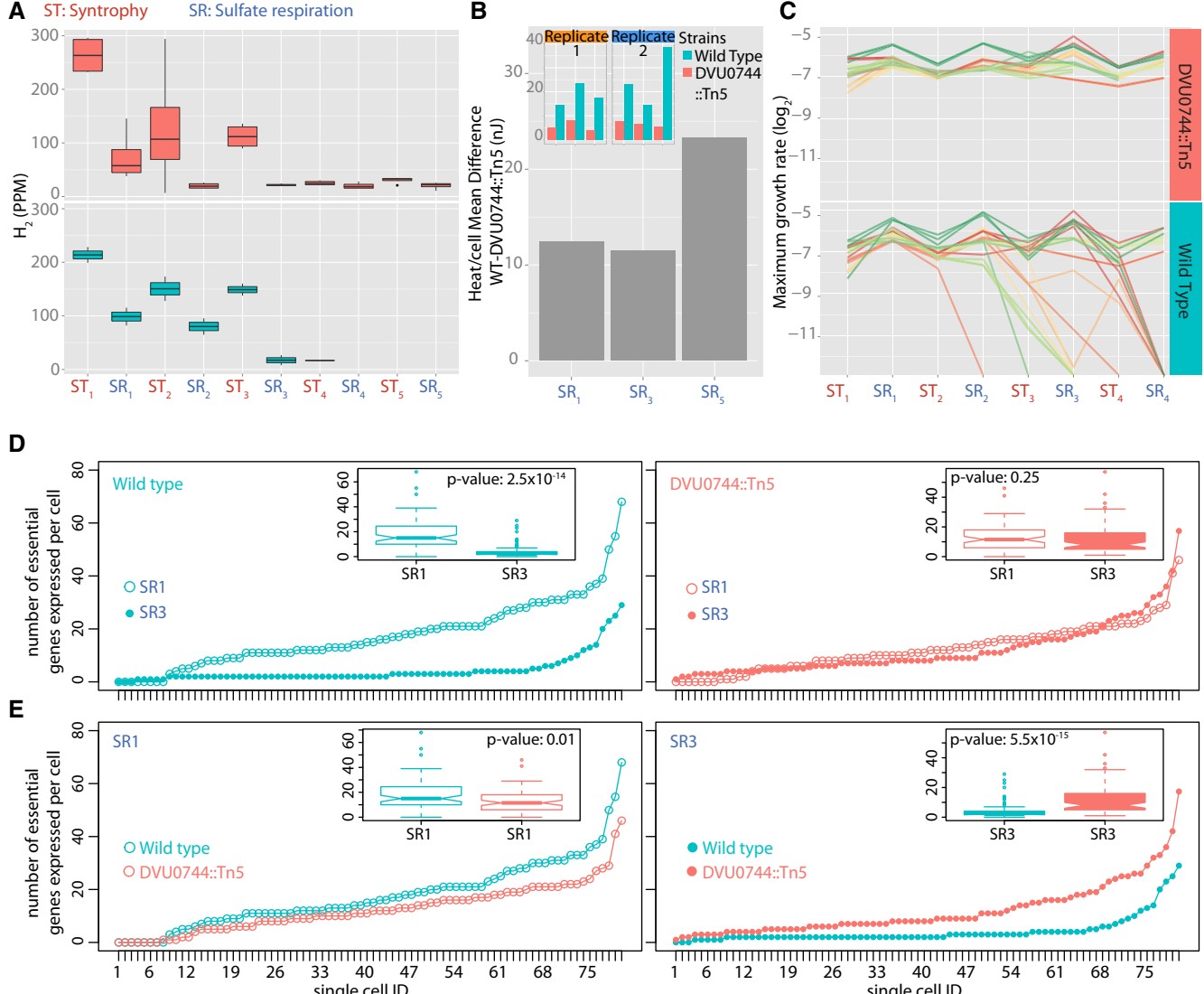

**Figure 6.  Evidence supporting the mechanism for microbial population collapse in a fluctuating environment.**

A   Boxplots for $H_2$ levels in replicate wild-type (lower panel) and mutant (upper panel) co-cultures.

B   Microcalorimetry measurement of heat released (nj) per cell upon transition of wild-type and mutant co-cultures into SR conditions. Inset plots report total heat released by wild-type (green) and mutant (red) co-cultures in SR1, SR3, and SR5 conditions, in two replicate experiments. Gray bar plots represent difference between average heat production per cell of wild-type and mutant strains, across the three successive SR conditions.

C   Growth rates for > 24 replicate co-cultures of Mm with either wild-type Dv (lower panel) or DVU0744::Tn5 (upper panel).

D   Significantly fewer essential genes are expressed across most single cells of the wild-type population that is unable to grow in the subsequent transition. Here, the number of essential genes expressed above background genomic DNA are plotted for each analyzed single cell and cells are ordered by rank abundance for WT SR1-initial transfer versus SR3 transfer prior to collapse (left); and mutant SR1- initial transfer versus SR3 where growth continues in contrast to the WT population (right). Insets are boxplots of the same data with *P*-values calculated from the equivalent of Mann–Whitney test (*n* = 80 single cells).

E   Number of essential genes expressed per cell in wild type is compared to mutant during SR1 (left) and SR3 (right). See Materials and Methods. Insets are boxplots of the same data with *P*-values calculated from the equivalent of Mann–Whitney test (*n* = 80 single cells).

Data information: (A, D and E) The lower and upper ends of the boxes ("hinges") correspond to the first and third quartiles (the 25th and 75th percentiles). Horizontal lines correspond to median values. Error bars extend from the upper or lower hinges to the highest or lowest values that are within 1.5× IQR (interquartile range) of the hinge, respectively. (D and E) The notches extend to ±1.58 IQR/sqrt.

Source data are available online for this figure.

the energy term for regulation since their genotypes differ by a single regulator (DVU0744). Hence, we reasoned that the difference in amount of heat released by the two strains can be attributed to the energy expended toward direct and indirect regulation of genes by DVU0744. We found significantly higher amount of heat per cell released by the wild type across both the early and late transition, relative to the regulatory mutant. This result was consistent with the prediction that conditional regulation in a rapidly fluctuating environment may present an energetic burden for maintaining relational resilience (Fig 6B; Table EV7).

### Replicates of wild-type co-cultures diverge and collapse at different times as a result of model-predicted dilution effect

The complete conditional repression of SR-essential genes in most cells, during ST conditions, would be expected to cause large "partitioning errors" at cell division in the wild type (Huh & Paulsson, 2011). The bottlenecking step when a small aliquot was transferred to a new growth condition could amplify this effect further and introduce greater variation across replicates of the wild-type co-culture. Indeed, variability in lactate consumption profiles of wild-type replicate co-cultures increased near the point of collapse (Fig EV3). The actual collapse potentially occurs when a critical fraction of the cells (it need not be all cells) in the transferred aliquot does not express at least one copy of one or more proteins essential for growth in a specific condition. If so, then the point of collapse would be predicted to emerge at variable times depending on the history of each replicate (Chen *et al*, 2014). Consistent with this predicted heterogeneity across replicates, collapse in wild-type lines occurred at variable times, mostly between the 3rd and 7th and a few at later transitions between SR and ST (Fig 6C). Notably, collapse events in four lines occurred under ST conditions (albeit significantly fewer than the number (16) that collapsed in SR conditions), demonstrating that this phenomenon is not restricted to just SR-associated functions and also affects ST-relevant proteins. Since DVU0744 targets seven transcription factors, indirect effects on global gene expression are expected and supported by the dysregulation of at least 93 ST-relevant genes in DVU0744::Tn5 (Fig 5A).

### Significantly fewer wild-type cells express essential transcripts prior to collapse

To obtain evidence for the mechanism for collapse, we quantified transcript abundance of 88 essential genes in samples of 80 single cells from wild-type and mutant cultures harvested from an early stage of experimental evolution (SR1) and a later stage (SR3) just prior to collapse of the wild-type line (Fig 6D and E; Tables EV8 and EV9). The fraction of single cells expressing essential transcripts dropped significantly in wild-type SR3 (before collapse), while the mutant cell population remained unchanged. It was also notable that, relative to the mutant, a larger fraction of wild-type single cells expressed a greater number of essential transcripts during SR1, and this profile was reversed in SR3—which explains the reversal of fitness ($f$) relationship between wild type and mutant in single ($f_{WT} > f_{mutant}$; Fig 2C) and multiple ($f_{mutant} > f_{WT}$; Fig 3) transitions (Fig 6E). These results provide definitive evidence for the mechanism of collapse and why it was rescued by a regulatory mutation. Expansion of the single-cell data and in-depth analysis

that is beyond the scope of this work is included as part of another manuscript (A.W. Thompson, S. Turkarslan, C.E. Arens, A. López García de Lomana, A.V. Raman, D.A. Stahl & N.S. Baliga, unpublished data).

## Discussion

During long-term experimental evolution and even in anaerobic digesters, a microbial culture or community experiences frequent, iterative changes and bottlenecks due to growth and successive transfers to new media (Demirel & Yenigün, 2002; Chen *et al*, 2008). We have shown that the conditional gene regulation required for adapting to such fluctuating environments can lead to population collapse. Although the modeling and experimentation demonstrates that the collapse was rescued by disruption of conditional regulation of essential transcripts and proteins in DVU0744:: Tn5, it does not rule out that it could have resulted from rescue of a similar dilution effect on other essential cellular components, such as ATP. In either case, the underlying mechanism for population collapse traces back to Tn5-insertion mediated disruption of DVU0744, a transcription regulator. The highly interconnected architecture of a GRN allows for a large number of disruptive mutations in regulators and noncoding regions to abolish conditional regulation and delay or even prevent this catastrophic phenomenon. Furthermore, validation of model predictions at an emergent phenotypic level (i.e. metabolic activity of Dv) accounts for the net consequence of dilution of one or as many as all essential components. While a previous study has shown how a single regulatory mutation can restore social independence (Fiegna *et al*, 2006), we have demonstrated how disrupted regulation stabilizes frequent transitions of a generalist between metabolic independence and mutualism, fostering its longer-term adaptive evolution in a fluctuating resource environment. In this study, availability of sulfate played a critical role in driving the wild-type co-culture to collapse as it determined the physiological mode of growth by driving large scale changes in gene expression. When sulfate is available, Dv preferentially grows via sulfate respiration by producing energy using sulfate as the final electron acceptor. However, in the absence of sulfate, the ability of Dv to ferment lactate is thermodynamically feasible only if the resulting hydrogen is consumed by its syntrophy partner, the methanogen. Our results demonstrate that when sulfate availability fluctuates too frequently in an environment where there is excess lactate and methanogen, the resulting gene regulation to shift repeatedly between SR and ST physiologies counterintuitively drives the community toward collapse.

Implications of this work extend to natural environments where organisms often experience fluctuating conditions and display relational resilience to establish stable interactions with the environment. The sensitivity of the regulatory network has presumably evolved to effectively maintain relational resilience of the community within a certain regularity and frequency of environmental change. Our results demonstrate that if conditional regulation in fluctuating resource environment interferes with intrinsic dynamics of restoring function in a microbial community (which is often driven by regulation), it may create conditions that lead to population collapse. Further, we provide evidence that the population

collapse occurs due to loss of relational resilience, that is, progressive drift from the environmentally relevant physiological state (Fuhrman *et al*, 2015). There are two interrelated reasons why this happens—first, conditional regulation drives dilution of key cellular components making the population ill-prepared to adapt to the next environmental change, and second, the energetic cost of restoring function becomes progressively more burdensome to a point that is unsustainable. This study also underscores how a systems biology approach for network identification and analysis can enable discovery of mechanisms that modulate the resilience of biological systems. Knowledge of such mechanistic underpinnings of microbial community resilience will be critical to understand ecosystem dynamics and engineer stable biotechnological processes (Song *et al*, 2015).

# Materials and Methods

## Strains and culture conditions

*Desulfovibrio vulgaris* Hildenborough wild type and mutants were obtained from Dr. Judy Wall (University of Missouri, Columbia, MO). All regulator mutants were obtained by transposon mutagenesis as described elsewhere (Fels *et al*, 2013). *Methanococcus maripaludis* S2 wild-type strain for co-culture experiments were obtained from Dr. John Leigh (University of Washington, Seattle, WA; Table EV12). Both mono- and co-culture strains were grown at 37°C. Co-cultures were established with wild-type Mm and either wild-type or regulatory mutants of Dv as described by Stolyar *et al* (2007). Media used in these studies were formulated based on CCMA medium as described previously (Walker *et al*, 2009). Lactate medium for ST growth contains 40 mM of lactate and no sulfate. Lactate–sulfate medium for SR growth contains 40 mM of lactate and 15 mM of sulfate. The latter formulation provided conditions such that Dv was entering into stationary phase when electron acceptor was depleted but electron and carbon source was still available (i.e. the fermentation state). Mono- and co-cultures were either grown in Balch tubes in 10-ml culture volume or in 200-ml serum bottles with 50-ml culture volume (for transcriptomic analysis). The headspace for tubes or cultures was filled with 80% $N_2$ and 20% $CO_2$ gas to create an anoxic environment. All media pHs were adjusted to 7.2 with bicarbonate.

## Variant discovery

Biomass sample collection for sequencing, DNA extraction, sequencing library preparation, and sequencing was performed as described before (Hillesland *et al*, 2014). To determine the mutations within each line, the resulting raw Illumina sequences were first quality controlled by using FastQC software (http://www.bioinformatics.babraham.ac.uk/projects/fastqc/). Sequences were then aligned to the reference *D. vulgaris* (NC_002937, NC_005863) and *M. maripaludis* (NC_005791) published genomes using breseq pipeline (Deatherage & Barrick, 2014). In addition, Genome Analysis Toolkit (GATK) pipeline (DePristo *et al*, 2011) for variation discovery was used as an additional validation. Briefly, first reads were aligned to reference genome by using bwa (Li & Durbin, 2009; -M -t 4 –R). Resulting alignment SAM files were converted to BAM files and

sorted. BAM files were marked for duplicates by using Picard Tools (http://picard.sourceforge.net/), and local realignment around indels was performed in order to identify most consistent placement of reads relative to the indels. Variant calling was performed by using either GATK UnifiedGenotyper or Varscan (Koboldt *et al*, 2012). Default parameters were used for UnifiedGenotyper while for Varscan parameters were –min-coverage 20 –min-var-freq 0.2. Resulting variants were annotated by using SnpEff tools (Cingolani *et al*, 2012).

## Growth assays

Growth of mono- and co-cultures was followed in replicate Balch tubes with 10 ml of culture volume under anaerobic conditions. Culture density was measured by tracking changes in optical density with a Thermo Scientific Spectronic 200 spectrophotometer at 600 nm wavelength. Maximum growth rate was determined as previously described (Turkarslan *et al*, 2011). Monocultures of Dv wild-type and regulatory mutants were cultivated in sulfate respiration growth medium until late stationary phase when sulfate was exhausted (lactate fermentation) for SR state transition experiments. This ensured depletion of electron acceptor but not carbon source due to medium formulation with limiting sulfate concentrations. At this stage, 0.5 ml of inoculum was transferred to 10 ml of fresh SR medium and growth was tracked. For state transition experiments with co-cultures alternating between ST and SR conditions, co-cultures were initially grown in ST growth medium until mid-log phase ($OD_{600}$ ~0.15) and 0.5 ml of inoculum was transferred into 10 ml of fresh SR growth medium. Cells were grown to early log density ($OD_{600}$ ~0.2), and 0.5 ml was transferred back into fresh ST growth condition medium. Alternating shifts between ST and SR conditions were continued as long as growth was observed.

## Dv custom tiling array construction

Whole-genome tiling arrays for *D. vulgaris* Hildenborough were designed with e-Array (Agilent Technologies), with strand-specific 60-mer probes and 149-bp spacing between adjacent probes for the main chromosome (NC_002937) and the Mega plasmid (NC_005863). Altogether the array contained a total of 60 K probes, including the manufacturer's controls. The microarrays were printed by Agilent Technologies. Labeling with Cyanine 3 (Cy3) and Cyanine 5 (Cy5) dyes (Molecular Probes and Kreatech BV), hybridization, and washing were performed as described earlier (Baliga *et al*, 2004). Arrays were scanned in ScanArray (Perkin-Elmer), and spot finding was done by Feature Extraction (Agilent Technologies). Normalization and statistical analysis were performed as described (Koide *et al*, 2009).

The microarray data reported in this paper have been deposited in the National Center for Biotechnology Information Gene Expression Omnibus (GEO) database (GEO accession no. GSE73105).

## Global gene expression profiling

Replicate co-cultures of wild-type and DVU0744::Tn5 mutant strains were grown anaerobically in 200-ml bottles with 50 ml culture media. The anoxic environment was created by filling the headspace with 80% $N_2$ and 20% $CO_2$ gas mixtures. Cells were grown in

ST medium to early log phase (OD$_{600}$ ~0.15), and 5 ml of culture was withdrawn into anaerobic Balch tubes. Another 0.5 ml of culture was transferred into SR medium, and another sample was harvested at mid-log phase (OD$_{600}$ ~0.3). The sampling schedule was repeated during SR growth and, subsequently, after transition back into ST.

Biomass was harvested anaerobically by centrifugation at 1,455 $g$ for 10 min; the cell pellet was immediately flash-frozen in liquid nitrogen and stored at −80°C. Total RNA was extracted with MasterPure Complete DNA and RNA Purification kit (Epicentre Technologies, Madison, WI), and DNA was removed using Ambion DNA-free kit (Life Technologies, Grand island, NY). Quality and purity of RNA samples was determined with Bioanalyzer, gel electrophoresis, and PCR. Sample preparation and hybridization was performed as described previously (Baliga *et al*, 2004).

### Transcript abundance determination with RNA-sequencing

Transitions, sample collection, RNA extraction, and DNA-cleanup were performed as described above. Illumina RiboZero rRNA removal kit (Illumina, San Diego, CA) was used for rRNA depletion. Samples were prepared with TrueSeq Stranded mRNA HT library preparation kit (Illumina, San Diego, CA) and sequenced on the NextSeq Sequencing machine in mid-output 150 v2 flow cell. Paired-end 75-bp reads were checked for technical artifacts using FastQC (Andrews, 2010) following Illumina default quality filtering steps. Reads were further trimmed for quality scores and cleaned up for adapter contamination with Trimmomatic (Bolger *et al*, 2014). Alignment of reads to reference was performed using STAR (Dobin *et al*, 2013) with modification of recommended parameters where appropriate. Read counts were collected by using HTSeq (Anders *et al*, 2015) followed by normalization and analysis with DESeq2 R package (Love *et al*, 2014). The analysis workflow, implemented as custom python and R scripts, is available at GitHub repository (https://github.com/sturkarslan/MSB-16-7058). All fastq files used in this study are deposited into SRA (accession number: GSE79022).

### Protein abundance determination with MS analysis

Cells were resuspended in 50 mM Tris, pH 8.0, 250 mM NaCl and subject to three rounds of freeze–thaw lysis. Extracted protein concentrations were measured using the BCA assay (Thermo Fisher Scientific). Equal protein amounts were denatured with 6 M urea, reduced with 5 mM dithiothreitol, alkylated with 25 mM iodoacetamide, and digested with Lys-C (1:200 w:w, 3 h, 37°C; Thermo Fisher Scientific). Urea was diluted to 1.5 M, and samples were further digested with trypsin overnight (1:25 w:w, 37°C; Thermo Fisher Scientific). Peptides were acidified with formic acid to stop digestion and purified using C18 reversed-phase chromatography (Nest Group). Purified peptides were separated by online nanoscale HPLC (EASY-nLC II; Proxeon) with a C18 reversed-phase column packed 25 cm (Magic C18 AQ 5 μm 100 A) over an increasing 120 min gradient of 5–35% Buffer B (100% acetonitrile, 0.1% formic acid) at a flow rate of 300 nl/min. Eluted peptides were analyzed with an Orbitrap Elite mass spectrometer (Thermo Fisher Scientific) operated in data dependent mode, with the Top20 most intense peptides per MS1 survey scan selected for MS2 fragmentation by rapid collision-induced dissociation (rCID; Michalski *et al*, 2012). MS1 survey scans were performed in the Orbitrap at a resolution of 240,000 at *m/z* 400 with charge state rejection enabled, while rCID MS2 was performed in the dual-linear ion trap with a minimum signal of 1,000. Dynamic exclusion was set to 15 s.

Raw output data files were analyzed using Maxquant (v1.5.5.1; Cox & Mann, 2008). Protein sequences of *Desulfovibrio vulgaris* (strain Hildenborough) and *Methanococcus maripaludis* (strain S2) were downloaded from UniProt (07-2016 release) and merged to create a single database. A reverse sequence database was used to impose a strict 1% FDR cutoff. Label-free quantification was performed using the MaxLFQ algorithm (Cox *et al*, 2014). Data processing was performed in Microsoft Excel and Perseus (v1.5.3.1; Tyanova *et al*, 2016). Contaminants, decoys, and single peptide identifications were removed. We required intensity values in two out of three replicates for at least one strain in the ST1+ST3 shifts, and the SR1SR3 shifts. Zero values in the remaining data were subsequently replaced by imputation.

We assessed the general trend for all 728 proteins by performing linear regression on protein quantification values along transition conditions. Linear regression was computed on MaxLFQ-normalized, transformed log$_2$ intensity values. We searched for proteins that were consistently and specifically downregulated in WT. We examined the following four constraints: (i) significant differences between ST1 and SR3 conditions (Student's *t*-test or Mann–Whitney *U*-test depending on normality of observed values; normality assessed by Shapiro–Wilk normality test), (ii) slope > twofold change decrease for WT profile, (iii) slope < twofold change decrease for DVU0744::Tn5 profile, and (iv) larger median for DVU0744::Tn5 than WT at SR3 conditions. We found 52 proteins that accommodated all four constraints. We performed a permutation test to confirm significance of this observation, that is, specific downregulation of proteins in DVU0744::Tn5 (*P*-value = 0.0025, Appendix Fig S2). Finally, we found 23 cases of proteins upregulated in WT but not in DVU0744::Tn5, a pattern far from expected by chance (permutation test, *P*-value < 1e-4) and only three cases of proteins downregulated specifically in DVU0744::Tn5 but not WT (permutation test, *P*-value < 1e-4). Code implementation is available at Github repository: https://github.com/sturkarslan/MSB-16-7058.

### EGRIN model

The EGRIN model was constructed from a compendium of transcriptome profiles for 3491 genes from 684 microarray experiments spanning 25 unique perturbations [MicrobesOnline (Dehal *et al*, 2009)]. Genomic data including annotations, sequences, and operon predictions were also downloaded from MicrobesOnline. The cMonkey algorithm was used to bicluster genes into conditionally co-regulated modules as described previously (Reiss *et al*, 2006; Bonneau *et al*, 2007). cMonkey identified 349 regulatory modules and 662 cis-regulatory motifs. 170 of 349 modules further passed our residual quality filter of 0.5 and 94 of 662 motifs passed *e*-value cut-off filter of 10. We applied the Inferelator (Bonneau *et al*, 2006) algorithm to infer 919 regulatory influences from 122 transcription factors and 12 environmental factors on 165 modules. Influence weight threshold of < −0.1 or > +0.1 was used to filter high confidence influences.

Predictive power of the EGRIN model was tested using an expression data set that was not used in model construction (Bonneau *et al*, 2006), and by comparison with manually curated regulon members in the RegPrecise (Novichkov *et al*, 2013). Significance of overlap between composition of EGRIN modules and RegPrecise regulons was calculated using a hypergeometric test followed by multiple hypothesis correction. The EGRIN model for Dv can be explored at http://networks.systemsbiology.net/syntrophy. R data file for EGRIN model and cytoscape session file are also available in Github repository: https://github.com/sturkarslan/MSB-16-7058.

### Gene/function enrichment analysis

Functional enrichment of GO terms within EGRIN modules was done with the TopGO R package (Alexa & Rahnenführer, 2009), using the hypergeometric test and multiple hypothesis testing (Benjamini & Hochberg, 1995). For each functional annotation, we surveyed functional assignments for the gene members of the regulatory modules. We identified enrichment of a specific functional class by calculating a hypergeometric $P$-value for each module and functional category pairs.

Genes that accumulated mutations during evolution experiments with co-cultures were investigated for functional enrichment by using DAVID Functional Annotation Tool (Huang *et al*, 2009). List of genes with mutations were analyzed against DAVID provided *Desulfovibrio vulgaris* Hildenborough background to highlight the most relevant functional annotation terms associated with these genes. This tool uses Fisher's exact test to determine gene enrichment in annotation terms and reports modified Fisher's exact $P$-values as described (Huang *et al*, 2009). Only terms with $P$-values smaller than 0.05 were included in the final analysis. Similar annotations were further clustered together by using Functional Annotation Clustering Tool from DAVID where appropriate. Degree of similarity between terms is measured by using the same techniques of Kappa Statistics, and clustering is performed by using fuzzy heuristic clustering as described (Huang *et al*, 2009). Cluster Enrichment Score is the geometric mean of member's $P$-values in a given Enrichment Cluster. Higher values are better. Only clusters with score bigger than 1.0 were included in the analysis.

Relative changes in gene expression were normalized to reference RNA from a mid-log phase co-culture of wild-type Dv with Mm. Differentially expressed genes were determined using Statistical Analysis of Microarrays (SAM) method with median false discovery set to zero genes. Differentially expressed genes for state transitions were identified by comparing early log phase expression ratios of first syntrophic state (ST1) to sulfate respiration state (SR) and sulfate respiration to second syntrophic state (ST2). For a given ST or SR growth condition, early log phase expression ratios were compared to mid-log phase expression ratios in order to determine differential expression changes during batch growth.

### Analytical methods

Hydrogen and methane were quantified with gas chromatography, organic acids, and alcohols by HPLC, and sulfate by ion chromatography, all as described previously (Stolyar *et al*, 2007; Walker *et al*, 2009).

### Cell counts

Determination of cell numbers for Dv and Mm was performed by flow cytometer and manual counts. Samples were collected at early log phase ($OD_{600}$ 0.15–0.20) before the transfer into the next growth conditions. Cultures were diluted proportionally by OD (1:1.5 for $OD_{600}$ 0.15, 1:2 for $OD_{600}$ 0.20) in lactate media and fixed in 0.37% formaldehyde for 5 min before flash-freezing in liquid nitrogen. Samples were thawed prior to time of counting, and relative ratios of Dv and Mm were obtained in disposable hemocytometers (InCyto C-Chip DHC-N01) under a Leica DM2000 microscope with a 40× objective. Total cell number was obtained by averaging the sum of both counts of Dv and Mm from five 0.004-$mm^3$ squares, in which 100–200 particles maximum were deposited.

For flow cytometric cell counts, fixed frozen cultures were thawed and diluted 1:20 in filtered lactate media. Flow cytometric cell counts were obtained with a BD Influx cell sorter (Becton Dickinson, Franklin Lakes, NJ) equipped with a small particle detector and 488-nm laser (Coherent, Santa Clara, CA). Sheath fluid was prepared from concentrated BioSure 8× Sheath fluid (BioSure, Grass Valley, CA) diluted to 1× concentration with DI water and filtered through a 0.2-µm pore size Sterivex filter (Millipore, Billerica, MA). Data collection was triggered on forward scatter (FSC). Dv and Mm cell populations were distinguished based on perpendicular and parallel FSC and side scatter (SSC) relative to standard 1-µm-diameter beads (Polysciences Inc., Warrington, PA) and referenced to pure monocultures. Volumes sampled were determined by weighing sample tubes before and after sampling on an analytical balance. Cell concentrations were calculated by dividing the number of events counted for each cell population by the volume sampled and providing the data as cell number per ml.

### Microcalorimetry

The microcalorimetric measurements were performed on a TAM III Nanocalorimeter (TA instruments, New Castle, USA), which measures the heat flow between a reaction cell and reference cell (Johansson & Wadsö, 1999; Wadso & Goldberg, 2001). Prior to each experiment, the heat flow response by the calorimeter was calibrated by electrical heating at 37°C with 4 ml Hastelloy reaction and reference cells containing 3 ml of the sterile lactate–sulfate medium described above. This electrical heating procedure was verified by measuring the heat of protonation of trishydroxymethylaminomethane (TRIS/THAM) at 25°C (Grenthe *et al*, 1970). After calibration, experimental cultures (wild-type and mutant DVU0744::Tn5) were inoculated as described above in Balch tubes containing lactate–sulfate medium. After a quick mixing, 3 ml of this culture was immediately dispensed into an autoclaved reaction cell in an anaerobic glovebox. The reaction cell was then transferred into the calibrated microcalorimeter. After the heat flow reached baseline, the reaction cell was removed and sampled for cell counts as described below. Heats of culture growth were derived by integration of the heat flow curves (Table EV7).

The calculated total heat was normalized to the total cell number in the vial. For normalizing the total heat released during batch growth in the microcalorimeter, the cell density was determined at the end of the experiment. 10 µl of 10% glutaraldehyde (Toumisis,

Rockville USA, stored at 4°C protected from light) was put into a sterile 1.2-ml cryovial (ThermoFisher Scientific, Waltham, USA). The reaction vials were manually shaken before taking 1 ml of culture and adding to the prepared cryovial. The cells were subsequently fixed for 10 min at room temperature in the dark. After shock-freezing, the samples were stored at −80°C until quantification by flow cytometry as described above.

**Single-cell transcript measurements**

Single-cell transcriptional changes for 94 Dv genes (Table EV8) were tracked across SR/ST transitions using protocols developed for the Fluidigm 96.96 Dynamic Array (Fluidigm Inc., South San Francisco, CA). Assays were chosen to include 88 SR-essential genes, 2 of which are highly expressed control genes. Transitions and sample collection were performed as described above. Assays were examined for the production of nonspecific products or cross-reactivity with other assays using melting curve analysis, resulting in the 94 high-quality assays used in the experiment. For each sample, we measured transcription of these 88 Dv genes in single cells with reverse transcription (RT single cells, $n = 80$), single cells without RT (noRT single cells, $n = 6$), positive internal controls of 10.6 pg extracted Dv RNA with ($n = 4$) and without ($n = 2$) RT treatment, and no template controls ($n = 4$). For cell sorting, Dv cells were distinguished from coexisting Mmp on the basis of side and forward scatter properties using a BD Biosciences Influx high-speed sorter with small particle detector as previously published (Thompson *et al*, 2015). Cells were sorted directly into a lysis/RT buffer solution consisting of 1× VILO Reaction Mix (Life Technologies), 6U SUPERase-In (Life Technologies), 0.5% NP-40 (Thermo Scientific), and nuclease-free water (TEKnova) in a 96-well plate format centrifuged, vortexed for 15 s, then frozen on dry ice, and stored at −80°C. Following cell lysis and RNA denaturation (90 s at 65°C), RT was carried out with 1× SuperScript Enzyme Mix (Life Technologies) and T4 Gene 32 Protein (New England BioLabs, Beverly, MA) by the following program: 25°C for 5 min, 50°C for 30 min, 55°C for 25 min, 60°C for 5 min, and 70°C for 10 min in a standard 96-well thermal cycler. Resulting cDNA was then amplified in a multiplexed specific target amplification (STA) reaction with all 94 Dv gene primer pairs (Table EV9) using TaqMan® PreAmp Master Mix (Applied Biosystems) and EDTA pH 8.0 by the following program; 95°C for 10 min, 25 cycles of 96°C for 5 s and 60°C for 4 min. STA-cDNA was then cleaned up by an Exonuclease I treatment (New England Biolabs, Beverly, MA). The resulting cDNA product was diluted fivefold in DNA Suspension Buffer (TEKnova), loaded into the Fluidigm 96.96 Dynamic Array following Fluidigm protocols (https://www.fluidigm.com/docume nts), and assayed against the 94 Dv assays by quantitative PCR using Sso Fast EvaGreen Supermix (Bio-Rad Laboratories) with ROX passive reference dye by the following program: 95°C for 60 s, 40 cycles of 96°C for 5 s and 60°C for 20 s, and melting curve from 60 to 95°C. Results were then analyzed with Fluidigm BioMark real-time qPCR Analysis software (Table EV9). Quality filtering and statistical tests to determine successful assays were performed as described in "Single cell transcription analysis" section (A.W. Thompson, S. Turkarslan, C.E. Arens, A. López García de Lomana, A.V. Raman, D.A. Stahl and N.S. Baliga, unpublished data).

**Single-cell transcription analysis**

BioMark Real-Time PCR Analysis software (Fludigm Inc. South San Francisco, CA) was used to view and analyze amplification and melting curves for each single cell and control for each of the 94 assays. Cycle of quantification ($C_q$) thresholds were set using the AutoGlobal method, and the baseline correction method used was Linear Derivative. Assays from cells or controls with atypical amplification curves were omitted from the analysis.

Additional quality control and analysis of single-cell data was performed in R. Assays from cells or controls with melting curve peaks ($T_m$) that were significantly different in a Student's *t*-test than the $T_m$ of the same assay applied to the positive internal controls were also omitted from analysis. No RT positive controls and no template controls were used to confirm that reactions were not contaminated, not producing nonspecific amplification products, and not cross-reacting with other assays. Relative quantity of molecules (RQ) was calculated from each $C_q$ value for more intuitive analysis (Ståhlberg *et al*, 2013). RQ $= 2^{(C_q\text{cutoff} - C_q)}$, with $C_q$cutoff set to the median of noRT single-cell controls ($n = 6$) across all assays so that the $C_q$ of a single cell with RT equal to the $C_q$ of a noRT single cell would yield an RQ of 1, or 1 molecule present in the reaction, which is what we expect for each single copy gene assayed in Dv and amplified from the noRT single-cell controls, where genomic DNA will be present. Assays from single cells that did not amplify ($C_q = $ NA) were set at RQ = 0.5, thus below the detection limit which is RQ = 1. Finally, RQ values for each assay and each RT single cell were compared to the distribution of RQs obtained from the noRT single cells using a Student's *t*-test. Assays from single cells with positive RQ values and *P*-value < 0.01 were concluded to express the target gene above the level of background genomic DNA. These data are represented in Fig 6D and E. RQ and $C_q$ values for each RT single cell and noRT single cell are presented in Table EV9.

**Phenomenological models of relative transcript and protein concentration changes per cell**

It has been suggested that in a given cell population, the distribution of protein abundance in single cells will follow the generalized gamma distribution, with the shape parameter corresponding to burst frequency, and the scale parameter corresponding to the amount of proteins produced per burst (Friedman *et al*, 2006).

The probability distribution function for protein copy number per cell is:

$$p(x) = \left( x^{a-1} \times e^{-x/b} \right)/(b^a \times \Gamma(a))$$

where *x* is the protein copy number in a cell, *a* is the transcriptional burst frequency per cell division, *b* is the amount of proteins produced per burst, which is related to translational efficiency, and *Γ* is the gamma function. This statistical model has been experimentally verified using the LacI repressor system (Cai *et al*, 2006). We simulated the protein copy number distribution in each phase of growth (ST and SR) for a gene that was conditionally repressed in ST, but essential for SR. This model was implemented by varying the starting parameters for burst frequency and size across ST and SR growth, as well as between the wild-type and mutant strains. To

account for the iterative transitions between SR and ST, the burst frequency was progressively modified by scaling it to the fraction of cells containing nonzero protein. We made this assumption on the basis that at least one copy of an essential protein would be necessary in order to initiate the transition to a new state. This is an analogy to the *lac* system, in which some lactose permease (the LacY gene product) is required in order to sense the presence of lactose, and begin the process of de-repression of the operon (Choi *et al*, 2008).

A relevant feature of this model is that it predicts that transcriptional repression of a gene by an environmental stimulus can shift a cell population from one in which most cells contain at least one copy of the regulated protein to one in which most cells contain no copies. Therefore, for our purposes, if the mutant strain has relaxed transcriptional repression of sulfate-essential genes due to deletion of DVU0744, then we may be observing an increase in cells that contain at least one copy of the regulated protein(s).

We simulated the protein copy number distribution in each phase of growth (ST and SR), assuming that a hypothetical gene was repressed in ST, but essential for SR. Counterintuitively, we also observed repression of some essential genes during SR, but not during ST. In either case, cycling of the regulation would lead to dilution of the protein abundance in a cell population during growth under repressing conditions and increase in protein when the repression was relieved in the other growth condition.

This model was implemented by varying the starting parameters for burst frequency and size across ST and SR growth, as well as between the wild-type and mutant strains. To account for the iterative transitions between the two growth conditions, the burst frequency was progressively modified by scaling it to the fraction of cells containing nonzero protein. We made this assumption on the basis that at least one copy of an essential protein would be necessary in order to initiate the transition to a new state. This is an analogy to the *lac* system, in which some lactose permease (the LacY gene product) is required in order to sense the presence of lactose, and begin the process of de-repression of the operon (Choi *et al*, 2008).

We observed large fluctuations in population-level repression depending on the gene, ranging from <20% to >60% repression in wild-type cells as compared to ST conditions. At the single-cell level, this can be equated to a reduction in transcriptional bursting frequency by a proportional amount. Loss of repression in the mutant was variable depending on the gene, but often times were nearly complete. In other words, we observed very little conditional repression in the mutant DVU0744::Tn5 strain for many sulfate-essential genes. Taken together, we simulated a range of increases in transcriptional bursting when repression was relieved. As starting conditions, we assumed the burst frequency under repression to be in the range of 0.2, based on the empirical fitting of Cai *et al* (2006). Under de-repression, the bursting frequency would increase to 2 bursts per cell cycle to simulate a 10-fold increase. In the mutant strain, the starting frequency during repression was set at 0.5 and the de-repressed frequency at 1.5, which simulates only a threefold difference to account for the observed loss of repression. We have simulated bursting frequencies in the range of 0.2 to 2, based on the assumption that most transcripts, including those for high abundance proteins, are often transcribed on the order of once per cell cycle, due to a limiting number of RNA polymerase enzymes (Bon *et al*, 2006).

We found with this set of parameters that the population of wild-type cells rapidly loses sulfate-essential proteins during repression in ST that is not fully restored during SR (Fig EV4A). The mutant strain, on the other hand, also experiences a decline in the fraction of cells with nonzero copies of the protein, but it is not as dramatic. To explore the range of this model, we varied the starting burst frequency, the amount of dysregulation, and the proteins per burst. When the starting burst frequency in the wild-type strain is increased to 0.4, indicating a fivefold repression instead of 10, the trajectories are similar but still show separation between genotypes (Fig EV4B). When the mutant burst frequency is adjusted to model less separation between ST and SR, the separation is even more dramatic, with the mutant strain showing a high proportion of cells with robust protein expression (Fig EV4C). With complete absence of conditional regulation, the trajectory follows a simple exponential decay with a stable baseline (Fig EV4D). Conversely, when the amount of protein produced per burst is increased, the effect is mitigated, and the wild-type strain can partially or completely rescue the dilution due to conditional repression (Fig EV4E and F). See figure legend for exact parameter values.

In sum, this model supports our hypothesis that differential conditional regulation may lead to a dominating effect of protein dilution in the wild-type cells that is not completely compensated for by de-repression when synthesis is permissive. This appears to be particularly true for low-abundance proteins with low output of proteins produced per burst and can be exaggerated given a more extreme loss of repression.

**Data availability**

The microarray data reported in this paper have been deposited in the NCBI GEO database (accession: GSE73105). RNA-sequencing data have been deposited in the NCBI GEO database (accession: GSE79022). Genome sequence data have been deposited into NCBI SRA (BioProject: PRJNA248017). Mass spectrometry data have been deposited in ProteomeXchange via MassIVE under the identifier PXD005456. Computational codes and EGRIN model are available at github (https://github.com/sturkarslan/MSB-16-7058) and Zenodo with identifier DOI: 10.5281/zenodo.197353.

**Expanded View** for this article is available online.

### Acknowledgements

This material by ENIGMA-Ecosystems and Networks Integrated with Genes and Molecular Assemblies (http://enigma.lbl.gov), a Scientific Focus Area Program at Lawrence Berkeley National Laboratory, is based upon work supported by the U.S. Department of Energy, Office of Science, Office of Biological & Environmental Research under contract number DE-AC02-05CH11231. JAR and MAG are supported by National Institute of General Medical Sciences Center for Systems Biology Award Number 2P50 GM076547. We thank Lisa Jones of the Fred Hutch Proteomics Resource for assistance with MS runs.

### Author contributions

ST designed, performed, and coordinated all experiments, developed, analyzed, and interpreted all data and models, and wrote manuscript. AVR developed the growth model. KLH helped with analysis of co-culture mutations and edited manuscript. CEA and SS helped with design and execution of experiments and edited manuscript. MAG and JAR performed proteomics

*Serdar Turkarslan et al*    Microbial collapse in variable environments    **Molecular Systems Biology**

experiments, MS analysis and helped editing the manuscript. DJR helped with EGRIN model construction. AWT performed flow cytometry analysis, designed and implemented the single-cell transcript analysis, and edited the manuscript. FvN and DG-L performed microcalorimetry and analyzed data. ALGL performed computational analysis for identifying dilution trends in transcript and protein abundances and helped editing the manuscript. GMZ tested and provided transposon mutants. JDW provided transposon mutant library for *Desulfovibrio vulgaris* Hildenborough and edited manuscript. DAS designed experiments, analyzed and interpreted all data, and edited manuscript. NSB conceived and coordinated the entire project, designed experiments, analyzed and interpreted data, wrote and edited manuscript.

## Conflict of interest

The authors declare that they have no conflict of interest.

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
