## [Review Process File · Molecular Systems Biology]

Mechanism for microbial population collapse in a fluctuating resource environment

Serdar Turkarslan, Arjun V Raman, Anne W Thompson, Christina E Arens, Mark A Gillespie, Frederick von Netzer, Kristina L Hillesland, Sergey Stolyar, Adrian López García de Lomana, David J Reiss, Drew Gorman-Lewis, Grant M Zane, Jeffrey A Ranish, Judy D Wall, David A Stahl and Nitin S Baliga

Corresponding author: Nitin Baliga, Institute for Systems Biology

Review timeline:

Submission date:	10 May 2016
Editorial Decision:	02 July 2016
Revision received:	22 December 2016
Editorial Decision:	22 January 2017
Revision received:	02 February 2017
Accepted:	06 February 2017

Editor: Thomas Lemberger

Transaction Report:

1st Editorial Decision

02 July 2016

Thank you again for submitting your work to Molecular Systems Biology. We have now finally heard back from the three referees who agreed to evaluate your manuscript. As you will see from the reports below, the referees find the topic of your study of potential interest. They raise, however, a series of important points, which should be convincingly addressed in a revision of the manuscript.

Without repeating all the comments listed in the reports below, the major points refer to the following aspects of the study:

- The method used to prioritize the three transcriptional regulators of SR should be clarified.
- The hypothesis that "loss of conditionally-repressed, essential proteins cause population collapse of the wild type co-culture" should be verified with protein-level measurements of some key (essential) components. Some of the proteins encoded by the 146 transcripts (including 8 essential genes) that display progressive dilution at the population level could be good examples.
- The validity of a significant correlation between transcript vs protein levels should be verified in Dv (Csardi et al refers to yeast, Van et al to an Archea) at least for key components. This seems particularly important to interpret the results obtained by quantifying 89 essential genes at the single cell level prior to collapse.
- While the microcalorimetric measurements represent a very interesting approach, they should be

put as much as possible in the context of the global cellular energetic budget to provide an indication of the relevance of these observations in explaining population collapse.

- The text could be clarified, especially in the Introduction and the beginning of Results. In particular it would be important to clearly delineate what was the outcome of your previous study and what is the starting point of the current study. We feel that this will help the reader to better understand the logic of the current work.
- The description of the construction of the Dv EGRIN model should be sufficiently detailed for readers to understand how it has been done. For example, it would be useful to provide details on which dataset were used ("739 microarrays spanning across 25 unique perturbations", which ones exactly?).
- The dataset used for the statistical validation of the network and the outcome of this validation ("the overlap between regulatory module members and RegPrecise regulon members") should also be formally reported.
- We appreciate that you provide a nice resource at <http://networks.systemsbiology.net/syntrophy/>. However, for the purpose of long term archival and reproducibility, we would kindly ask you to make available as Expanded View Model the Dv EGRIN model that includes "919 regulatory influences from 122 transcription factors, 346 "down-regulating" and 573 "up-regulating"
- To help readers finding the datasets associated with this study, please include the accession numbers and relevant links in a "Data availability" section at the end of Materials & Methods.

REFEREE REPORTS

Reviewer #1:

This study focuses on the behavior of an assembled microbial consortia in a fluctuating environment where the biological interactions between community members change over time. The communities consist of *Desulfovibrio vulgaris* (wild-type and three regulatory mutants) and *Methanococcus maripaludis* to a. This fluctuating environment consisted in switches between a condition in which *D. vulgaris* is metabolically independent (i.e. it performs sulfate respiration) and a condition in which both species grow through a syntrophic interaction. The surprising outcome of these experiments is that the wild type consortia tend to collapse in a fluctuating environment while one of the regulatory mutants sustained a stable population. The authors explored the possible mechanisms behind these observations by modeling and experimental work.

The found the manuscript very difficult to understand, and this makes it also challenging to review it. I will first comment on where I had difficulties with understanding, and provide some other suggestions for modifications.

- The main problem I had was that the connection between the previously published evolution experiment of obligate syntrophy (Hillesland & Stahl 2010 and Hillesland et al. 2014) and the current study of co-cultures in fluctuating environments was not clear to me. After reading the manuscript again I (think) I understood that the previously published study provided strains that were then analyzed in the present study, and that the analysis of these strains then pointed towards mutations whose effects were investigated here in fluctuating environments. I suggest to completely rewrite the introduction and the results and discussion to make the logical structure of the project easier to understand.

- also, it was not easy for me to understand how the three transcriptional regulators that the authors investigated were chosen (bottom of p. 6). Where these the only transcriptional regulators that the EGRIN analysis revealed? The manuscript provides very little details about the identification of the regulators (in general) and about the selection of these three (in particular). Furthermore, the text is hard to follow because it jumps from writing about the "three transcriptional regulators" (DVU0744,

DVU2275, DVU2802) to talking about three "TF mutants". I suggest to introduce these three TF mutants by saying that these TF mutants were obtained by transposon mutagenesis (i.e. disruptive mutations).

- another major comment: One of the main conclusions of the authors is that "the collapse [of the assembled consortium in fluctuating conditions] was caused by conditional gene regulation. It is not clear to me that one can draw this conclusion - as far as I can see the statement is based on the exclusion of an alternative hypothesis (competitive replacement of one of the two types), but there seem to be other alternative scenarios that that not been tested and excluded. For example, how do we know that the population collapse is not a consequence of an evolutionary change in one of the two partners? I think such a scenario is consistent with the observation that a deletion in a regulatory (which might result in the inactivation of a pathway that could, if mutated, lead to collapse) prevents collapse.

Other comments and suggestions:

- (related to the first comment above): at the beginning of the 'Results and Discussion' it did not become clear to me that the authors are writing about the previously published evolution experiment; instead, I understood this section as referring to the populations evolved under fluctuating environments, as described e.g. in the abstract.

- the authors seem to assume that most of the mutations that were identified by analyzing the previously published evolved populations have a phenotypic effect; for example, they refer to some of these mutations as "disruptive mutations in the GRN for SR". However, as far as I can see there is no support for this interpretation (except for the three regulators that are analyzed in more detail). I think it is also conceivable (although admittedly maybe not very likely) that a substantial fraction of these mutations might be phenotypically neutral.

In conclusion, I think the study addresses an interesting and novel topic, but I found it difficult to understand, and I did not get convinced that the main conclusions are well supported by the findings.

Reviewer #2:

This manuscript combined a systems biology approach to generate a model of gene regulation in *Desulfovibrio vulgaris*, and then use that model to investigate observations that had been made on organismal responses to transient environments. The combination of bioinformatics analysis with experimental analyses is noteworthy and meritorious.

Specific comments"

1. P 11, l 14-17: I see a flaw in your logic here. The citations pertain to "steady-state" conditions, whereas yours were not. At best, they might be "balanced growth" conditions, but even this is unclear under the transitions you applied to the cultures. I appreciate the difficulty in measuring single-cell protein abundances, but you might be more circumspect regarding this aspect. Your data certainly show significant changes in transcriptional response.

2. P13, l 4-8 and Fig. 6B: This calorimetry data needs some context. Clearly there is a difference in heat release, but is it significant in the context of the cell's energy budget? You argue that there are additional energetic demands in the wild type as it regulates gene expression after transition (and implicitly this creates a negative selection pressure vs. the mutant). However, it seems to me that this depends upon what this cost is relative to the total energy budget of growth. Figure 6B is not so clear - do the inset bars represent total heat output for the experiment? (the y-axis legend mentions difference, but this makes no sense - please clarify). If so, and there were similar amounts of growth in WT v mutant, this implies the energetics were a strong selective force.

Reviewer #3:

In this paper, Turkaslan et co-workers investigated the mechanism behind the collapse of microbial populations when they are submitted to fluctuating environments. Their study model was *Desulfovibrio vulgaris*, a microbe which can grow independently in the presence of sulfate or in syntrophic association (obligate mutualism) with *Methanococcus maripaludis* when sulfate is depleted of the media.

In their previous works, the group evolve *Desulfovibrio vulgaris* populations to be highly specialized in mutualistic association with *Methanococcus maripaludis* through evolution of 1000 generations. They noticed the high incidence of mutations in regulatory genes as well as intergenic regions and, additionally, the decrease of the sulfate reduction capacity of *Desulfovibrio vulgaris*. Here, the authors used RNA-seq analysis, microcalorimetry, and single cell transcriptome analysis to demonstrate the causes of the collapse phenomenon once *Desulfovibrio vulgaris* populations were submitted to fluctuating environments (presence or absence of sulfate). According to their findings, the conditional regulation of genes necessary for adaptation of microbes to diversified environments can also drive the collapse of a population submitted to intense fluctuations in its environment, mainly due to a combination of progressive dilution of key cellular components (transcripts and proteins) and the increased energetic cost of restoring function after every change. Additionally, the authors demonstrated that the collapse phenomenon was avoided by a single mutation in a regulatory gene.

The paper is interesting, it is an example of synergy between systems network modeling and microbial experimentation. The writing is good enough, although in many cases convoluted and unnecessarily complex (but much less so than other papers from this group). Some comments that need to be addressed:

1. The basic hypothesis behind the mechanism of population collapse is at the protein level, but any proteomics are strikingly missing. This is an obvious omission for this work and it needs to be corrected through proteomics - even if this is western blots for the key components to validate the mRNA-protein abundance assumption. (Major point)
2. Quantify in a cost-benefit model the fitness cost for generalist/specialist populations. Also the group uses microcalorimetry to demonstrate energetic burden of wild type populations after environmental fluctuations. This is an interesting technique but to my knowledge there is no clear benchmark with other methods, including those that directly measure fitness or energy-related chemical compounds. The authors must provide this link and measure the energetic burden with another method, comparing the results (Major point)
3. I would like the group to investigate if this statement is correct in the microbes they are working on: "However, because variation in steady state abundance of most proteins can be explained by corresponding changes in mRNA levels (Csárdi et al, 2015; Van et al, 2008)". This is a controversial point and I expect that it will hold in a case-by-case basis.
4. In the last paragraph of page 6, using EGRIN authors identify three novel transcriptional regulators, including DVU0744 (a repressor with 128 target genes), DVU2275 (an activator with 240 target genes), and DVU2802 (a repressor with 119 target genes). Please report how many target genes are common here (venn diagram). Figure 2A indicates that these transcriptional regulators have several common target genes. Similar to DVU0744, DVU2802 is also a repressor, however it is not clear why cells with DVU2802 collapse, please elaborate.
5. A comprehensive analysis of target genes of these three transcriptional regulators can further narrow down the space of suspected genes responsible for the collapse and possibly can explain the trend of collapse observed in Figure 3.
6. It is not clear if the concentrations of sulfate and lactate play a role with the collapse. Please provide convincing arguments and it would help if a suppl. figure on this subject is included.

Summary of response

We were very encouraged by the enthusiasm of all three reviewers and greatly appreciate their constructive critique. We have addressed their critique by conducting additional analyses and also new experiments including mass spectrometry-based quantitative proteomics to track abundance changes in 728 proteins during laboratory evolution. The new efforts have generated additional evidence that regulation in a highly fluctuating environment leads to dilution of essential cellular components, leading to collapse of a microbial community. The new data demonstrates that essential proteins are progressively diluted with each transition into syntrophy and complements the observation of the same phenomenon at the transcript level. Notably, this dilution effect was observed only in the wild type background, and not in the DVU0744::Tn5 background, implicating regulation by this TF as the root cause of the dilution effect. Results from these new experiments are presented in the **Results** section on page 13 lines 3-13, with an accompanying **Figure 5E**, supplementary files (**Table EV10**, **Figure EV5**) and corresponding Methods section on page 21 L4-23 and page 22 L1-8.

In addition, we have addressed all of the other critiques by improving the presentation of data and discussion, and performing additional statistical analyses, such as comparing relationship between RNA and protein abundance under non-steady state levels. The new work has added one new Figure panel, one Expanded View Figure file, one Expanded View Table, five Data Source files for figures and has helped to improve clarity of the manuscript per reviewer guidance. We also created a public GitHub repository to archive codes and models and connected it with Zenodo service for acquiring a DOI number for long-term archiving purposes. Below, we provide a point-by-point response to each editorial and reviewer requests and critique:

Editorial requests

REQUEST 1: *The method used to prioritize the three transcriptional regulators of SR should be clarified.*

RESPONSE: Done –see Response to Reviewer 1 CRITIQUE 2.

REQUEST 2: *The hypothesis that "loss of conditionally-repressed, essential proteins cause population collapse of the wild type co-culture" should be verified with protein-level measurements of some key (essential) components. Some of the proteins encoded by the 146 transcripts (including 8 essential genes) that display progressive dilution at the population level could be good examples.*

RESPONSE: Done –see Response to Reviewer 3 CRITIQUE 1.

REQUEST 3: *The validity of a significant correlation between transcript vs protein levels should be verified in Dv (Csardi et al refers to yeast, Van et al to an Archea) at least for key components. This seems particularly important to interpret the results obtained by quantifying 89 essential genes at the single cell level prior to collapse.*

RESPONSE: Done –see Response to Reviewer 2 CRITIQUE 1 and Reviewer 3 CRITIQUE 1.

REQUEST 4: *While the microcalorimetric measurements represent a very interesting approach, they should be put as much as possible in the context of the global cellular energetic budget to provide an indication of the relevance of these observations in explaining population collapse.*

RESPONSE: Done –see Response to Reviewer 2 CRITIQUE 2 and Reviewer 3 CRITIQUE 2.

REQUEST 5: *The text could be clarified, especially in the Introduction and the beginning of Results. In particular it would be important to clearly delineate what was the outcome of your previous study and what is the starting point of the current study. We feel that this will help the reader to better understand the logic of the current work.*

RESPONSE: Done –see Response to Reviewer 1 CRITIQUE 1.

REQUEST 6: *The description of the construction of the Dv EGRIN model should be sufficiently detailed for readers to understand how it has been done. For example, it would be useful to provide details on which dataset were used ("739 microarrays spanning across 25 unique perturbations", which ones exactly?).*

RESPONSE: Done –We added Expanded View Table EV11 that lists details for all the datasets used in model reconstruction.

REQUEST 7: *The dataset used for the statistical validation of the network and the outcome of this validation ("the overlap between regulatory module members and RegPrecise regulon members") should also be formally reported.*

RESPONSE: Done –see Expanded View Table EV4. Since submission of this manuscript, a new version of RegPrecise (v4.0) has been released. We re-analyzed our network model by using new RegPrecise regulon membership data. We updated Expanded View Table EV4 with new analysis results and also added columns indicating regulatory module memberships, RegPrecise regulon membership and overlap between the two.

REQUEST 8: *We appreciate that you provide a nice resource at <http://networks.systemsbio.net/syntrophy/>. However, for the purpose of long term archival and reproducibility, we would kindly ask you to make available as Expanded View Model the Dv EGRIN model that includes "919 regulatory influences from 122 transcription factors, 346 "down-regulating" and 573 "up-regulating"."*

RESPONSE: Done –In order to make EGRIN model available we provide in two different format. First, EGRIN model environment can be explored by using RData file (Expanded View Model EV2) that can be directly loaded into R. Second, we provide Cytoscape file (Expanded View Model EV1) of the EGRIN model that can be opened and explored in Cytoscape. For long term archiving purposes both models are stored in a GitHub repository (<https://github.com/sturkarslan/MSB-16-7058>). Models in GitHub are also archived in Zenodo with accession id DOI: 10.5281/zenodo.197353

REQUEST 9: *To help readers finding the datasets associated with this study, please include the accession numbers and relevant links in a "Data availability" section at the end of Materials & Methods.*

RESPONSE: Done –We have added “Data availability” section at the end of Materials & Methods with appropriate accession numbers and links to repositories.

REQUEST 10: *On a more editorial level, we would kindly attract your attention the few following general points related to data and figure presentation:*

- *As you may have noticed, we recently replaced Supplementary Figures by Expanded View Figures (EV, see examples in <http://msb.embopress.org/content/11/6/812>). In this format, a limited number of Supplementary Figures (max 5) can be integrated in the article as EV figures that are interactively collapsible/expandable and will be typeset by the publisher. In this case, the figures should be cited as 'Figure EV1, Figure EV2' etc... in the text and their respective legends should be added to the main text after the legends of regular figures. The illustrations should be provided as separate files.*

RESPONSE: Done –We have updated manuscript to replace Supplementary Figures with Expanded View figures. These figures are cited as Figure EV1 through Figure EV5 in the text. We also appended “Expanded View Figure Legends” section after the legends for regular figures.

- *For the figures that you do NOT wish to display as Expanded View figures items, they should be bundled together with their legends in a 'traditional' supplementary PDF, now called the *Appendix*. Appendix should start with a short Table of Content and the figures should be named and referred to in the main text as: "Appendix Figure S1, Appendix Figure S2" etc. See detailed instructions regarding expanded view here: <http://msb.embopress.org/authorguide#expandedview>.*

RESPONSE: Done –We do not have additional figures for bundling in Appendix but Supplementary Methods now named “Appendix Supplementary Methods”.

- *Additional Tables/Datasets should be labeled and referred to as Table (or Dataset) EV1 etc. Table/Dataset legends can be provided in a separate tab in case of .xls files. Alternatively, you can upload a .zip file containing the Table/Dataset file and a separate README .txt file with the legend/description.*

RESPONSE: Done –We have updated our Supplementary Tables (xls) to label and refer to them as Table EV1 through Table EV11. Table legends are provided in a separate tab labelled “Legend” inside each .xls file.

- *We would also encourage you to include the *source data for figure panels* that show essential data, so that readers can download these data directly from the figure. Source data files are associated to individual panels of main figures. *Numerical data* should be provided as individual .xls files (including a tab describing the data) or csv or tab-delimited text files. *For 'blots' or microscopy*, uncropped images should be submitted. For *network visualization*, Cytoscape session files, if available, can be supplied. The files should be labeled as "Source Data for Figure 1A" etc. Source Data for Expanded View and Appendix figures should be uploaded as a single ZIP file containing all the Source Data for Expanded View and Appendix content. (Additional information on source data is available in the "Guide for Authors" section at <http://msb.embopress.org/authorguide#sourceData>)*

RESPONSE: Done –We included new source data files for Figures 2-6. Source Data for Figure 1 is available in Table EV2, Source Data for Figure 5E is available in Table EV10 and Source Data for Figure 6DE is available in Table EV9.

Reviewer #1:

*This study focuses on the behavior of an assembled microbial consortia in a fluctuating environment where the biological interactions between community members change over time. The communities consist of *Desulfovibrio vulgaris* (wild-type and three regulatory mutants) and *Methanococcus maripaludis* to a. This fluctuating environment consisted in switches between a condition in which *D. vulgaris* is metabolically independent (i.e. it performs sulfate respiration) and a condition in which both species grow through a syntrophic interaction. The surprising outcome of these experiments is that the wild type consortia tend to collapse in a fluctuating environment while one of the regulatory mutants sustained a stable population. The authors explored the possible mechanisms behind these observations by modeling and experimental work.*

The found the manuscript very difficult to understand, and this makes it also challenging to review it. I will first comment on where I had difficulties with understanding, and provide some other suggestions for modifications.

CRITIQUE 1: *The main problem I had was that the connection between the previously published evolution experiment of obligate syntrophy (Hillesland & Stahl 2010 and Hillesland et al. 2014) and the current study of co-cultures in fluctuating environments was not clear to me. After reading the manuscript again I (think) I understood that the previously published study provided strains that were then analyzed in the present study, and that the analysis of these strains then pointed towards mutations whose effects were investigated here in fluctuating environments. I suggest to completely rewrite the introduction and the results and discussion to make the logical structure of the project easier to understand.*

RESPONSE: The reviewer has misinterpreted how the two prior studies connect to the current study. We have revised the manuscript to clarify this point, and provide a brief explanation below for the benefit of the reviewer.

The work of Hillesland & Stahl 2010 and Hillesland et al. 2014 investigated physiological and genomic changes across *Desulfovibrio vulgaris* Hildenborough (Dv) and *Methanococcus maripaludis* (Mm) when the two organisms were required to co-evolve in an obligately interdependent syntrophic association. The major finding from this prior work was that the obligate association with Mm resulted in erosion of sulfate respiration functions, thereby compromising the capability of Dv to live independently if excess sulfate were to become available in the future. Interestingly, the prior studies revealed that mutations had also accumulated at high frequency in components of the regulatory network –a finding that raised interesting questions regarding the role of regulation in evolution. The current study investigates this question in great depth –this is the

link between the prior studies and this study. In brief, the current study was motivated and guided by a hypothesis formulated from observations made in the previous studies --that regulation is important only when the environment fluctuates. Accordingly, we expected that regulation of sulfate respiration would be essential if the two organism community was required to sustain *facultative* syntrophy –i.e., ability to grow in changing conditions under which the community has to frequently shift between growing with sulfate respiration and growing in syntrophic association. Inference of the regulatory network, characterization of regulatory mutants; laboratory evolution in fluctuating environmental conditions; extensive molecular, transcriptomic, proteomic, and single cell characterization of why the wild type community collapsed, and why the DVU0744 regulatory mutant did not collapse –are all unique aspects of this study.

We have rewritten the text to make this easier to understand, improved the presentation of information and data, clarifying connections to previous studies and improve overall readability. These sections are marked with red text color (P3, L21-23; P4, L12-23; P5, L1-3).

CRITIQUE 2: - *also, it was not easy for me to understand how the three transcriptional regulators that the authors investigated were chosen (bottom of p. 6). Where these the only transcriptional regulators that the EGRIN analysis revealed? The manuscript provides very little details about the identification of the regulators (in general) and about the selection of these three (in particular). Furthermore, the text is hard to follow because it jumps from writing about the "three transcriptional regulators" (DVU0744, DVU2275, DVU2802) to talking about three "TF mutants". I suggest to introduce these three TF mutants by saying that these TF mutants were obtained by transposon mutagenesis (i.e. disruptive mutations).*

RESPONSE: We have clarified how regulators of sulfate respiration were discovered and how the mutants were obtained. In brief, regulators of sulfate respiration in DvH had not previously been identified. For this reason, we had to first decipher the gene regulatory network by using an established systems biology approach. Briefly, we compiled a compendium of transcriptomes from studies that had probed the transcriptional responses of DvH across diverse environmental conditions. We performed biclustering to identify conditionally co-regulated gene modules (biclusters) using the cMonkey algorithm. Next, we used a regression-based approach (Inferelator) to infer the putative transcription factors (TFs) and environmental factors that could be implicated in regulation of the co-regulated gene modules. DVU0744, DVU2275, DVU2802 were the top three TFs that were implicated in the regulation of modules that were enriched for genes associated with sulfate respiration functions. We characterized the predicted roles of these three TFs by assaying sulfate-respiration-relevant fitness defects of three mutants, each with a transposon-insertion in one of the three TFs. We have added verbiage to page 7 lines 3-11 to explain our selection methodology. In addition, per Reviewer 1's request we introduce TF mutants in the appropriate section (P7, L3-11).

CRITIQUE 3- *another major comment: One of the main conclusions of the authors is that "the collapse [of the assembled consortium in fluctuating conditions] was caused by conditional gene regulation. It is not clear to me that one can draw this conclusion - as far as I can see the statement is based on the exclusion of an alternative hypothesis (competitive replacement of one of the two types), but there seem to be other alternative scenarios that that not been tested and excluded. For example, how do we know that the population collapse is not a consequence of an evolutionary change in one of the two partners? I think such a scenario is consistent with the observation that a deletion in a regulatory (which might result in the inactivation of a pathway that could, if mutated, lead to collapse) prevents collapse.*

RESPONSE: We disagree with the reviewer. We have considered many plausible mechanisms for the collapse phenomenon, and have provided thoughtful discussion and experimental evidence to rule out alternate hypotheses. For instance, we conducted flow cytometry and hemacytometer cell counting to rule out the hypothesis that the microbial community might have collapsed because Mm cells were diluted out in the co-culture with wild type Dv. However, our conclusion is not just based on exclusion of hypotheses; rather, we conducted numerous experiments to gather evidence for underlying differences across Mm co-cultures with wild type and DVU0744::Tn5 that might explain why one collapsed and the other didn't. Expectedly, we observed large scale differences in the numbers of genes that were differentially regulated across the two sets of the evolution lines. Specifically, we observed that significantly fewer genes were differentially regulated in the TF mutant as compared to the wild type –an observation that was also supported by

microcalorimetry, which showed that by regulating more genes the wild type generated twice as much heat relative to the TF mutant during transition between sulfate respiration and syntrophy. Using an established model developed by Xie et al we predicted the consequence of this striking difference in scale of gene regulation. The model predicted that products (transcripts and proteins) of regulated genes will get diluted with every transfer into a non-permissive growth condition (i.e., a condition in which a given gene is repressed), and this dilution effect will be dampened and rescued upon disrupting the regulatory network. We *tested* the model prediction at a transcriptional level using RNA-seq and also by doing targeted qRT-PCR on 89 essential genes across 80 single cells. Both sets of experiments unequivocally demonstrated that there was precipitous dilution of transcripts in the wild type background, and this effect was rescued in the TF mutant. While this evidence was compelling, two reviewers requested that we should also demonstrate the phenomenon at a protein level. We have now performed mass spectrometry-based quantitative proteomics to demonstrate that indeed the dilution phenomenon is also observable at the protein level.

Importantly, our data and observations refute both of the two alternate hypotheses suggested by Reviewer 1: **Reviewer Hypothesis 1**: an evolutionary change in one of the partners could have caused collapse; and **Reviewer Hypothesis 2**: mutation in an essential pathway in the wild type caused collapse, the regulatory mutation (DVU0744::Tn5) disrupted the network that activates this pathway and prevented collapse. There is overwhelming evidence against both of these hypotheses.

Evidence against Reviewer Hypothesis 1: This hypothesis can be refuted by a very simple explanation. If an evolutionary change were the underlying cause for collapse then we should have observed similar frequency of collapse events across all genetic backgrounds, wild type and mutant. This was not the case. On the contrary, the collapse was observed reproducibly in the wild type background, but never in the DVU0744:Tn5 background. The dynamics of collapse were also highly reproducible in the other two TF mutant backgrounds, occurring in DVU2802::Tn5 with similar dynamics as the wild type, and occurring in an intermediate timeframe in the DVU2275::Tn5 background (longer than wild type but shorter than DVU0744::Tn5). So the collapse phenomenon is characteristic of the genetic background, and NOT a result of a random evolutionary change in one of the two interacting organisms.

Evidence against Reviewer Hypothesis 2: The reviewer's second hypothesis is that a deleterious mutation in an essential pathway in the wild type caused collapse. And that this phenomenon was rescued by the DVU0744::Tn5 mutation, which somehow disrupted pathway activation. This is a convoluted and flawed hypothesis that is counter to the actual observation. The collapse was observed in a wild-type background, in which all regulatory functions were intact. We do not understand how a deleterious mutation in an essential pathway would be selected in the first place, let alone how it would lead to sudden collapse. If the reviewer meant to say that the regulatory mutant is somehow defective in activation of a deleterious pathway, then this should be a strong pressure for selection of spontaneous mutants that disrupt the regulatory network in the wild type background. However, across nearly 30 evolution lines of wild type co-cultures we did not observe selection of a single mutant that was able to rescue the collapse phenomenon.

In summary, we have provided overwhelming evidence for the underlying mechanism for collapse of the wild type co-culture, and an explanation for why this phenomenon was rescued in the regulatory mutant. The other two reviewers have made a note of this and have commented specifically on the merits of our computational and experimental strategies that have demonstrated the underlying cause for collapse. In fact, we have generated even more evidence for the underlying mechanism as a response to specific requests from the other two reviewers.

Other comments and suggestions:

CRITIQUE 4 - *(related to the first comment above): at the beginning of the 'Results and Discussion' it did not become clear to me that the authors are writing about the previously published evolution experiment; instead, I understood this section as referring to the populations evolved under fluctuating environments, as described e.g. in the abstract.*

RESPONSE: The reviewer is correct in their interpretation. The previously published work laid the groundwork for performing laboratory evolution of the Dv and Mm co-cultures. **However, the design and implementation of laboratory evolution experiments under fluctuating environmental conditions are all new and were done exclusively as part of this study.** In order to clarify what has been done in the previous work we edited Introduction, P4 L12-23 and the beginning of the Results and Discussion section (P5, L1-3).

CRITIQUE 5: - *the authors seem to assume that most of the mutations that were identified by analyzing the previously published evolved populations have a phenotypic effect; for example, they refer to some of these mutations as "disruptive mutations in the GRN for SR". However, as far as I can see there is no support for this interpretation (except for the three regulators that are analyzed in more detail). I think it is also conceivable (although admittedly maybe not very likely) that a substantial fraction of these mutations might be phenotypically neutral.*

RESPONSE: We agree with the reviewer that not all mutations have deleterious phenotypic effects. We were very careful and conservative in calling mutations deleterious. Only indels that introduce frameshifts or premature stop codons in coding sequences were labeled "disruptive mutations" in the manuscript.

Reviewer #2:

*This manuscript combined a systems biology approach to generate a model of gene regulation in *Desulfovibrio vulgaris*, and then use that model to investigate observations that had been made on organismal responses to transient environments. The combination of bioinformatics analysis with experimental analyses is noteworthy and meritorious.*

Specific comments:

CRITIQUE 1: *P 11, l 14-17: I see a flaw in your logic here. The citations pertain to "steady-state" conditions, whereas yours were not. At best, they might be "balanced growth" conditions, but even this is unclear under the transitions you applied to the cultures. I appreciate the difficulty in measuring single-cell protein abundances, but you might be more circumspect regarding this aspect. Your data certainly show significant changes in transcriptional response.*

RESPONSE: We thank Reviewer 2 for bringing this point to our attention. Reviewer 2 is correct: there are no studies investigating mRNA-protein correlations under non-steady state conditions. We have clarified this in the manuscript. However, this point is secondary to the main thesis of our argument that the collapse occurred because of dilution of essential cellular components, including transcripts and proteins. Nonetheless, we have now also performed global shotgun proteomics of co-cultures of Mm with the wild type and regulatory mutant strains of Dv at varying stages of shifts between sulfate respiration and syntrophic growth states (See P13, L3-13). There were many cases where changes in the protein level correlated significantly with transcript level changes; however, the degree of correlation was not always high. However, this experiment directly verified the model-predicted dilution effect for many essential proteins (Figure 5E, Figure EV5 and Table EV10). Importantly, dilution of proteins was observed almost exclusively in the wild type background, and not in the DVU0744::Tn5. We have revised the manuscript to clarify the point on mRNA-protein relationship and present new evidence for the model prediction that transcriptional regulation in a frequently fluctuating environment leads to precipitous dilution of regulated gene products (transcripts AND proteins), and given that the function of these proteins is essential this is the most likely cause for collapse of the co-culture.

CRITIQUE 2: *P13, l 4-8 and Fig. 6B: This calorimetry data needs some context. Clearly there is a difference in heat release, but is it significant in the context of the cell's energy budget? You argue that there are additional energetic demands in the wild type as it regulates gene expression after transition (and implicitly this creates a negative selection pressure vs. the mutant). However, it seems to me that this depends upon what this cost is relative to the total energy budget of growth. Figure 6B is not so clear - do the inset bars represent total heat output for the experiment? (the y-axis legend mentions difference, but this makes no sense - please clarify). If so, and there were similar amounts of growth in WT v mutant, this implies the energetics were a strong selective force.*

RESPONSE: It is very difficult to dissect the total energy budget of a cell into individual energy terms such as energy for regulation, maintenance, growth etc. Therefore, we used total amount of heat released as a proxy for total energy expenditure of cells in each culture, and attributed the difference in heat produced by wild type vs mutant to lack of regulation in the latter. Our rationale was that both strains would have comparable energy terms for growth, maintenance etc. and that the only difference would be in the energy term for regulation since their genotypes differ by a single

regulator (DVU0744). Hence, we reasoned that the difference in amount of heat released by the two strains can be attributed to the energy expended towards direct and indirect regulation of genes by DVU0744.

Inset bars in Figure 6B show total heat/cell that is released by each replicate culture upon transition to a new growth condition. The main barplot, on the other hand, highlights the difference in amount of heat released by wild type vs. mutant. Height of barplots in the main chart represents the difference between average amount of heat released by wild type cultures AND average amount of heat released by mutant cultures upon transition and growth in a new culture condition (data are shown for 1st, 3rd and 5th transition from syntrophy to sulfate respiration). The difference in heat released is indeed *very significant* and is equivalent to $\sim 1/3^{\text{rd}}$ the total amount of heat released by wild type cultures (see inset barplot for reference). We clarified this point in the figure legend. Please also see our response (Response 3) to Reviewer #3's CRITIQUE 2 for providing context for microcalorimetry data.

Reviewer #3:

In this paper, Turkaslan et co-workers investigated the mechanism behind the collapse of microbial populations when they are submitted to fluctuating environments. Their study model was Desulfovibrio vulgaris, a microbe which can grow independently in the presence of sulfate or in syntrophic association (obligate mutualism) with Methanococcus maripaludis when sulfate is depleted of the media.

In their previous works, the group evolve Desulfovibrio vulgaris populations to be highly specialized in mutualistic association with Methanococcus maripaludis through evolution of 1000 generations. They noticed the high incidence of mutations in regulatory genes as well as intergenic regions and, additionally, the decrease of the sulfate reduction capacity of Desulfovibrio vulgaris. Here, the authors used RNA-seq analysis, microcalorimetry, and single cell transcriptome analysis to demonstrate the causes of the collapse phenomenon once Desulfovibrio vulgaris populations were submitted to fluctuating environments (presence or absence of sulfate). According to their findings, the conditional regulation of genes necessary for adaptation of microbes to diversified environments can also drive the collapse of a population submitted to intense fluctuations in its environment, mainly due to a combination of progressive dilution of key cellular components (transcripts and proteins) and the increased energetic cost of restoring function after every change. Additionally, the authors demonstrated that the collapse phenomenon was avoided by a single mutation in a regulatory gene.

The paper is interesting, it is an example of synergy between systems network modeling and microbial experimentation. The writing is good enough, although in many cases convoluted and unnecessarily complex (but much less so than other papers from this group). Some comments that need to be addressed:

CRITIQUE 1: *The basic hypothesis behind the mechanism of population collapse is at the protein level, but any proteomics are strikingly missing. This is an obvious omission for this work and it needs to be corrected through proteomics - even if this is western blots for the key components to validate the mRNA-protein abundance assumption. (Major point)*

RESPONSE: We agree with the reviewer's comment that demonstration of the dilution effect at the protein level would strengthen the main conclusions of this manuscript. In response to the reviewer's request we have conducted a new set of experiments to complement transcriptomics measurements and gather evidence for the "dilution effect" at the proteome level. In brief, we have performed global, quantitative shotgun proteomics using an Orbitrap Elite mass spectrometer to quantify abundance changes in 728 proteins during transitions of Mm co-cultures with wild type and DVU0744::Tn5 between sulfate respiration and syntrophy. The quantitative proteomics analysis demonstrated that 52 proteins, including several that are essential, were progressively diluted in wild type Dv during transitions, but not in the regulatory mutant (p-value: 0.0025). Notably, an insignificant number of proteins (3 proteins) displayed a reverse trend, i.e., they were diluted in the mutant but not in the wild type –demonstrating unequivocally that regulation by DVU0744 in response to frequent transitions between sulfate respiration and syntrophy leads to progressive dilution of both transcripts and proteins. Results from these new experiments are presented in

Results on page 13 lines 3-13, with an accompanying **Figure 5E**, and supplementary files (**Table EV10, Figure EV5**).

CRITIQUE 2: *Quantify in a cost-benefit model the fitness cost for generalist/specialist populations. Also the group uses microcalorimetry to demonstrate energetic burden of wild type populations after environmental fluctuations. This is an interesting technique but to my knowledge there is no clear benchmark with other methods, including those that directly measure fitness or energy-related chemical compounds. The authors must provide this link and measure the energetic burden with another method, comparing the results (Major point)*

RESPONSE: Microcalorimetric measurements reflects sum of both physical and chemical processes in the cell. This makes it very challenging to dissect the total energy budget of a cell into individual energy terms such as energy for regulation, maintenance, growth etc. Therefore, we used total amount of heat released as a proxy for total energy expenditure of cells in a culture. Our rationale was that both strains would have comparable energy terms for growth, maintenance etc. and that the only difference would be in the energy term for regulation since their genotypes differ by a single regulator (DVU0744). Further, we reasoned that the difference in amount of heat released by the two strains can be attributed to the energy expended towards direct and indirect regulation of genes by DVU0744.

The energetic burden of regulation is described by the relationship of total heat released to the total cell counts. The cell counts in themselves are an indicator for fitness, and the heat is linked to the Gibbs energy consumed and enthalpy of growth (von Stockar *et al*, 2006; von Stockar & Liu, 1999). High heat values to a low cell number thus indicate a decreased fitness for the wild type as the high heat indicates a high enthalpy in relation to low growth. According to the second law of thermodynamics this enthalpy must be related to energy consumption which did not translate into cell growth, hence the increased ratio of heat per cell for the wild type.

The sensitivity and accuracy of microcalorimeters makes them ideal for this application as they have been used widely and effectively to assay bioenergetics of diverse metabolic processes, including sulfate reduction, denitrification, fermentation, and even aging. Here are few examples:

- In *Desulfovibrio* species, microcalorimetry was used to assay how chromium altered bioenergetics during growth, and even identify the specific phase of growth that was associated with reduction activity (Chardin *et al*, 2002).
- In terms of correlation of microcalorimetric measurements with energy-related chemical compounds, Pamatmat *et al*. showed strong correlation between heat production, ATP concentration and electron transport activity in marine sediments (Pamatmat *et al*, 1981).
- Braeckman *et al*, 2002 exploited sensitivity of microcalorimetry to assay metabolic rates in aging *C. elegans* (Braeckman *et al*, 2002).
- Braissant *et al*., used microcalorimetry to track heat flow during growth on *E. coli* on M9 medium and they were able to deconvolute peaks in heat production to glucose respiration, glucose fermentation and lactose fermentation (Braissant *et al*, 2010).

It is important to emphasize that all of these studies establish the use of microcalorimetry to detect subtle differences in heat production due to variations in bioenergetics of cellular metabolism and growth. We have demonstrated how comparative microcalorimetry analysis of two strains that differ in a single regulator can provide insights into the relative burden of regulation. Further studies with more complex experiment designs (e.g., comparing difference in heat flow upon pulsing different concentrations of sulfate into sulfate-limited cultures of wild type and mutant strains, or doing the same experiment with a strain in which the regulator is expressed from an inducible promoter) will be required to further dissect the individual energy terms –but this is beyond the scope of this manuscript.

CRITIQUE 3: *I would like the group to investigate if this statement is correct in the microbes they are working on: "However, because variation in steady state abundance of most proteins can be explained by corresponding changes in mRNA levels (Csárdi *et al*, 2015; Van *et al*, 2008)". This is a controversial point and I expect that it will hold in a case-by-case basis.*

RESPONSE: There is one study that has used a multiple regression approach to investigate correlation between variations in transcript and protein concentrations in *Desulfovibrio* species (Nie *et al.*, 2006). This study revealed that variation in mRNA abundance alone could explain just 20-28% of total variation in protein concentration. The authors reported that factors that influenced this mRNA-protein relationship included expected properties such as protein stability, and unexpected factors such as the functional role of proteins. Specifically, they showed that there was significantly higher mRNA-protein correlation for genes involved in energy-metabolism, a result that was reproduced by comparative analysis of data generated through our newly performed proteomics experiments. We observed that changes in 171 proteins out of 728 proteins identified were correlated to variations in their transcript levels (Pearson correlation coefficient cutoff: 0.5). These genes were involved in metal-binding (30), protein biosynthesis (10), iron-sulfur binding (11), energy production and conversion (9) and signaling (21) (DAVID Functional enrichment analysis). The previous statement regarding mRNA-protein correlation has been appropriately revised in the manuscript.

Most importantly, while the proteomics analysis enabled assessment of the mRNA-protein relationship issue, the main objective of this analysis was to directly test the model prediction that regulation in a fluctuating environment drives dilution of protein levels. The new results are consistent with the model prediction providing further evidence for the mechanism underlying the collapse phenomenon.

CRITIQUE 4: In the last paragraph of page 6, using *EGRIN* authors identify three novel transcriptional regulators, including DVU0744 (a repressor with 128 target genes), DVU2275 (an activator with 240 target genes), and DVU2802 (a repressor with 119 target genes). Please report how many target genes are common here (venn diagram). Figure 2A indicates that these transcriptional regulators have several common target genes. Similar to DVU0744, DVU2802 is also a repressor, however it is not clear why cells with DVU2802 collapse, please elaborate.

RESPONSE: According to *EGRIN*, the venn diagram illustrates numbers of targets that are unique and shared across the three transcription factors. It is not just the number of common targets of these transcription factors, but also the context for their regulation that might explain why DVU2802::Tn5 behaves like the wild type and collapses after few transitions, but DVU2275 and DVU0744 are able to sustain a greater number of transitions. The earlier version of the *EGRIN* model used in this study does not provide context for regulation, although a newer version of the model has the potential to provide that level of information (assuming the compendium of transcriptomes include sufficient experiments for a given environmental context. This is something we will follow up on in the future, but is beyond the scope of the study at this time.

CRITIQUE 5: A comprehensive analysis of target genes of these three transcriptional regulators can further narrow down the space of suspected genes responsible for the collapse and possibly can explain the trend of collapse observed in Figure 3.

Answer: Per the reviewer's request we have performed functional enrichment analysis for shared and unique targets of the three transcriptional regulators. The shared targets were enriched for ATP synthesis, ion transport, oxidative phosphorylation and cell inner membrane. DVU0744 and DVU2802 targets were enriched for rRNA binding and ribosome functional terms, but these functions were absent among the DVU2275 targets. On the other hand, DVU2275 genes that are not targeted by DVU2802 or DVU0744 were enriched for transport and cell membrane related terms. Functional enrichment analysis can be a useful tool in order to connect a group of genes with similar biological functions. However, redundancy of different ontologies and missing annotations are potential problems with such analyses and partially explains why the enrichment analysis described above did not implicate a particular function or process as the underlying cause for collapse. A more relevant analysis that we reported in the manuscript is that a significant number of differentially regulated genes during

transitions were determined in an independent study as being essential for growth with sulfate respiration (**Figure 5A**). Moreover, we also demonstrated through transcriptomics and now also with proteomics that because of regulation a number of essential genes suffer dilution at the transcript and protein levels in the wild type. The single cell transcriptomics results were perhaps the most compelling in this regard as it showed that dilution is a global effect in the wild type that is completely absent in the DVU0744::Tn5 background. So the collapse phenomenon is not something that could be attributed to one or few genes or functions, rather it is a systems effect of regulation.

CRITIQUE 6: *It is not clear if the concentrations of sulfate and lactate play a role with the collapse. Please provide convincing arguments and it would help if a suppl. figure on this subject is included.*

RESPONSE: Sulfate concentration does play a critical role in driving the wild type co-culture to collapse as it determines the physiological mode of growth by driving large scale changes in gene expression. When available Dv preferentially grows via sulfate respiration by producing energy using sulfate as the final electron acceptor. However, in the absence of sulfate, its ability to ferment lactate is thermodynamically feasible only if the resulting hydrogen is consumed by its syntrophy partner, the methanogen. Thus, availability of sulfate, lactate, and proximity to a methanogen all play a role in determining whether *Desulfovibrio* grows via sulfate respiration or syntrophy, by appropriately turning on or off the expression of relevant genes for each physiology. The design of the laboratory evolution essentially probes the role of regulation in conferring resilience to the Dv community when it has to switch between these two modes of growth in response to frequent fluctuations in sulfate availability. The conclusion is that when sulfate availability fluctuates too frequently in an environment where there is excess lactate and methanogen, the resulting gene regulation to adapt to the fluctuations in sulfate concentration counterintuitively drives the community towards collapse.

References

- Braeckman BP, Houthoofd K, De Vreese A & Vanfleteren JR (2002) Assaying metabolic activity in ageing *Caenorhabditis elegans*. *Mech. Ageing Dev.* **123**: 105–119
- Braissant O, Wirz D, GÄ¶pfert B & Daniels AU (2010) Use of isothermal microcalorimetry to monitor microbial activities. *FEMS Microbiol. Lett.* **303**: 1–8 Available at: <http://femsle.oxfordjournals.org/cgi/doi/10.1111/j.1574-6968.2009.01819.x> [Accessed November 13, 2016]
- Chardin B, A. D, F. C, M. F, P. G & M. B (2002) Bioremediation of chromate: thermodynamic analysis of the effects of Cr(VI) on sulfate-reducing bacteria. *Appl. Microbiol. Biotechnol.* **60**: 352–360 Available at: <http://link.springer.com/10.1007/s00253-002-1091-8> [Accessed November 13, 2016]
- Nie L, Wu G & Zhang W (2006) Correlation of mRNA expression and protein abundance affected by multiple sequence features related to translational efficiency in *Desulfovibrio vulgaris*: a quantitative analysis. *Genetics* **174**: 2229–43 Available at: <http://www.ncbi.nlm.nih.gov/pubmed/17028312> [Accessed November 13, 2016]
- Pamatmat MM, Graf G, Bengtsson W & Novack CS (1981) Heat Production, ATP Concentration and Electron Transport Activity of Marine Sediments. *Mar. Ecol. Prog. Ser.* **4**: 135–143
- von Stockar U & Liu J-S (1999) Does microbial life always feed on negative entropy? Thermodynamic analysis of microbial growth. *Biochim. Biophys. Acta - Bioenerg.* **1412**: 191–211
- von Stockar U, Maskow T, Liu J, Marison IW & Patiño R (2006) Thermodynamics of microbial growth and metabolism: An analysis of the current situation. *J. Biotechnol.* **121**: 517–533

Thank you again for submitting your work to Molecular Systems Biology. We have now heard back from the referees who accepted to evaluate the revised study. As you will see, the referees are now mostly satisfied with the modifications made. I am therefore pleased to inform you that we will be able to accept your manuscript for publication pending the following minor amendments:

- Please include a short explanation of the impact of sulfate/lactate in the text, as requested by reviewer #2
- Supplementary information includes mostly Methods. If you are not opposed, we would prefer to have a unique Materials & Methods section in the main paper where you merge the information currently provided in Supplementary information.
- The Yeast strains should then become Table EV12 and 'Supplementary Experimental Procedures Fig 1' should become Figure EV6 and referenced in the text.
- The Appendix could then in principle be removed or reduced to the remainder, if any. If some form of the Appendix is kept, please include a Table of Content.

REFEREE REPORTS

Reviewer #2:

I had reviewed a previous version of this manuscript. I am satisfied that the authors have addressed my concerns, and my opinion is that they have also addressed the concerns of the other reviewers.

Reviewer #3:

All remarks addressed, most of them in a satisfactory manner. The authors performed the proteomics experiments that now show some evidence of their claims (point 1). They haven't provided a cost-benefit model, still an issue for me, but explain their point of view for the calorimetric methods they used (point 2). The analysis on point 3 shows that their previous claim was mistaken and they have revised their wording. I still have an issue with the usefulness of the EGRIN analysis (point 4); the authors completed the Venn Diagram and tried to explain it through functional enrichment but it didn't give any signal - they attribute it to being a "systems effect of regulation", which still should show up although noise in annotations may have impacted the results. Finally they explain the impact of sulfate/lactate and they should add it in the manuscript.

2nd Revision - authors' response

02 February 2017

Responses to editorial and reviewer requests

- **Editorial/Reviewer Request 1:** Please include a short explanation of the impact of sulfate/lactate in the text, as requested by reviewer #2
 - **Response:** We integrated the impact of sulfate/lactate on the collapse of co-cultures as they experience alternating shifts between sulfate respiration (in the presence of sulfate and excess lactate) and syntrophic (in the absence of sulfate and excess lactate) conditions in the manuscript on Page 17, L2-11 (highlighted in red)
- **Editorial Request 2:** - Supplementary information includes mostly Methods. If you are not opposed, we would prefer to have a unique Materials & Methods section in the main paper where you merge the information currently provided in Supplementary information.
 - **Response:** We expanded the Methods section in the main paper and removed Supplementary Methods.

- **Editorial Request 3:** - The Yeast strains should then become Table EV12 and 'Supplementary Experimental Procedures Fig 1' should become Figure EV6 and referenced in the text.
 - **Response:**
 - Strains table in the Supplementary Methods is now presented as Table EV12.
 - 'Supplementary Experimental Procedures Fig 1' is also moved out of Supplementary Methods and presented as Figure EV6.
 - In addition, table in the Supplementary methods displaying the model parameters is converted into Table EV13.

- **Editorial Request 4:** The Appendix could then in principle be removed or reduced to the remainder, if any. If some form of the Appendix is kept, please include a Table of Content.

Response: We removed the Appendix section after combining Methods section, and creating Table EV12, Table EV13 and Figure EV6.

3rd Editorial Decision

06 February 2017

Thank you again for sending us your revised manuscript. We are now satisfied with the modifications made and I am pleased to inform you that your paper has been accepted for publication.

Corresponding Author Name: Nitin S Baliga

Manuscript Number: MSB-16-7058